

# Simulated ocean oxygenation during the interglacials MIS 5e and MIS 9e

Bartholomé Duboc[1,2,3], Katrin J. Meissner[1,3], Laurie Menviel[1,4], Nicholas K.H. Yeung[1,3],
Babette Hoogakker[5], Tilo Ziehn[6], and Matthew Chamberlain[7]

[1]Climate Change Research Centre, University of New South Wales, Sydney, NSW, Australia
[2]CentraleSupelec, Gif-sur-Yvette, France
[3]ARC Centre of Excellence for Climate Extremes, University of New South Wales, Sydney, NSW, Australia
[4]The Australian Centre for Excellence in Antarctic Science, University of New South Wales, Sydney, NSW, Australia
[5]The Lyell Centre, Heriot-Watt University, Edinburgh, UK
[6]Oceans and Atmosphere, CSIRO, Aspendale, VIC, Australia
[7]Oceans and Atmosphere, CSIRO, Hobart, TAS, Australia

**Correspondence:** Katrin J. Meissner (k.meissner@unsw.edu.au)

**Abstract.**

Recent studies investigating future warming scenarios have shown that the ocean oxygen content will continue to decrease over the coming century due to ocean warming and changes in oceanic circulation. However, significant uncertainties remain regarding the magnitude and patterns of future ocean deoxygenation. Here, we simulate ocean oxygenation with the ACCESS
ESM1.5 model during two past interglacials that were warmer than the preindustrial climate, the Last Interglacial (Marine Isotope Stage (MIS) 5e, ∼129–115 ka) and MIS 9e (∼336-321 ka). While orbital parameters were similar during MIS 5e and MIS 9e, with lower precession, higher eccentricity and higher obliquity than pre-industrial, greenhouse gas radiative forcing was highest during MIS 9e. We find that the global ocean is overall less oxygenated in the MIS 5e and MIS 9e simulations compared to the preindustrial control run and that oxygen concentrations are more sensitive to changes in the distribution of
incoming solar radiation than to differences in greenhouse gas concentrations. Large regions in the Mediterranean Sea are hypoxic in the MIS 5e simulation, and to a lesser extent in the MIS 9e simulation, due to an intensification and expansion of the African Monsoon, enhanced river run-off and resulting freshening of surface waters and stratification. Upwelling zones off the coast of North America and North Africa are weaker in both simulations compared to the preindustrial control run, leading to less primary productivity and export production. Antarctic Bottom Water is less oxygenated, while North Atlantic Deep
Water and the North Pacific Ocean at intermediate depths are higher in oxygen content. All changes in oxygen concentrations are primarily caused by changes in ocean circulation and export production and secondarily by changes in temperature and solubility.

## 1 Introduction

The global ocean oxygen inventory has decreased by over 2% since 1960 and the volume of anoxic waters has more than
quadrupled over the same time period (Schmidtko et al., 2017; Levin, 2018). This decrease was caused by a decline in ocean



solubility due to higher ocean temperatures (Bopp et al., 2013), but changes in ocean circulation and biological consumption also contributed to the observed changes (Ito et al., 2017; Breitburg et al., 2018). Model simulations project that this decline will continue into the future, although the spatial patterns and magnitude are model-dependant and strongly influenced by the models' ocean diffusivity parameters (Duteil and Oschlies, 2011; Gnanadesikan et al., 2012; Bopp et al., 2013; Long et al.,
2016; Bahl et al., 2019; Frölicher et al., 2020; Kwiatkowski et al., 2020; Chamberlain et al., 2024). Future ocean deoxygenation could eventually pose severe problems for ocean ecosystems and human societies (Diaz and Rosenberg, 2008; Vaquer-Sunyer and Duarte, 2008; Sampaio et al., 2021; Penn and Deutsch, 2022; Santana-Falcón et al., 2023) and there has been a growing interest in recent years to better quantify ocean deoxygenation and understand its drivers.

Ocean oxygenation has also changed in the past. For example, Scholz et al. (2014) find sulphidic conditions in the near-
surface sediments of the Peruvian upwelling area during the Last Interglacial period (LIG) (Marine Isotope Stage (MIS) 5e, ~129-116 ka ago), based on sedimentary molybdenum accumulation. Records of sedimentary $\delta^{15}N$ and redox-sensitive metals show that interglacials in the late Quaternary led to an expansion and/or intensification of near-surface and intermediate depth suboxic zones in the eastern Pacific margins and the Arabian Sea compared to glacials (Ganeshram et al., 2000; Galbraith et al., 2004; Nameroff et al., 2004; Glock et al., 2022). These changes in oxygenation have been generally assigned to changes
in oxygen supply to the global thermocline (Galbraith et al., 2004; Meissner et al., 2005; Muratli et al., 2010; Jaccard and Galbraith, 2012). For large regions of the deep ocean this trend was inverted (Hoogakker et al., 2018), and deep and bottom waters were better ventilated during interglacials compared to glacials. Jaccard et al. (2009) report close to suboxic conditions during glacial intervals of the past 150 ka compared to better ventilated conditions during MIS 5e and the Holocene (the past 11,700 years) in the deep subarctic Pacific. In the deep equatorial Pacific there is multi-proxy evidence for reduced oxygen
concentrations in all deep Pacific Ocean water masses below 1 km during glacials compared to MIS 5e and the Holocene (Anderson et al., 2019; Jacobel et al., 2020). Bottom waters in the northern Arabian Sea, on the Portuguese margin and in the North and South Atlantic, and South Atlantic Circumpolar Deep Water (CDW), were also less oxygenated during glacial episodes compared to MIS 5e and the Holocene (den Dulk et al., 1998; Hoogakker et al., 2015, 2016; Lu et al., 2016; Gottschalk et al., 2020). It is noteworthy that most reconstructions of ocean oxygenation in the late Quaternary compare glacials with
interglacials; there is very little evidence on how oxygen concentrations varied between interglacials. This is an important caveat, as better understanding of ocean oxygen variability in the past, and in particular, ocean oxygenation during past warm episodes, could inform on the main processes influencing current ocean deoxygenation.

There is also evidence of intervals with severe anoxia in the Mediterranean Sea during past interglacials (Sachs and Repeta, 1999; Rohling et al., 2015; Rush et al., 2019). These intervals are characterised by sediment layers with elevated organic carbon
concentrations (sapropels) and coincide with astronomically timed episodes of monsoon intensification, causing enhanced runoff from North Africa into the Mediterranean Sea and leading to stratification (Rohling et al., 2015). The resulting anoxia was often more intensely developed in the eastern Mediterranean than in the western Mediterranean.

Here, we analyse simulations integrated with the Australian Earth System Model ACCESS-ESM1.5 for two interglacials, MIS 5e (LIG) and MIS 9e. The LIG is an interesting time period because it was globally the warmest interglacial of the
past 800 ka (Past Interglacials Working Group, PAGES, 2016). Being the most recent interglacial, the spatial and temporal



resolution of climate proxies is also much better than for earlier interglacials. The LIG climate has been recently simulated by a variety of climate models under the Paleoclimate Model Intercomparison Project 4 (PMIP4) lig127k experiment (Otto-Bliesner et al., 2017, 2021). MIS 9e (∼336-321 ka ago) stands out by its high greenhouse gas forcing, with the highest carbon dioxide and methane concentrations over the past 800 ka (Past Interglacials Working Group, PAGES, 2016). It is also one of

the three interglacials (together with MIS 11 and MIS 17) with the lowest value of seawater $\delta^{18}$O, which might imply high sea levels and small continental ice volumes (Elderfield et al., 2012; Past Interglacials Working Group, PAGES, 2016), although sea-level records from the Red Sea find that MIS 5e had higher sea levels than MIS 9e (Rohling et al., 2009). MIS 9e was an exceptionally short interglacial with the warmest temperatures in Antarctica over the past 800 ka (Past Interglacials Working Group, PAGES, 2016). We will focus on how changes in orbital parameters and greenhouse gas concentrations influenced

simulated ocean oxygenation for these two interglacials.

## 2 Methods

We use the Australian Community Climate and Earth System Simulator Earth System Model, ACCESS-ESM1.5 (Ziehn et al., 2020) to integrate three equilibrium simulations under preindustrial, MIS 5e and MIS 9e boundary conditions. The model includes the atmosphere UK Met Office Unified Model (UM) version 7.3 (Martin et al., 2010; The HadGEM2 Development

Team, 2011), the Community Atmosphere Biosphere Land Exchange model (CABLE) version 2.4 (Kowalczyk et al., 2013), the NOAA/GFDL Modular Ocean Model (MOM) version 5 (Griffies, 2014), and the Los Alamos National Laboratory sea ice model (LANL CICE) version 4.1 (Hunke et al., 2010). The components communicate with each other through the Ocean Atmosphere Sea Ice Soil - Model Coupling Toolkit (OASIS-MCT, Craig et al. (2017)). The land and atmosphere components have a horizontal resolution of 1.875°x1.25°, with 38 vertical levels for the atmosphere model. The ocean and sea ice com-

ponents have a resolution of 1°x1°, with 50 vertical levels for the ocean model. The horizontal resolution of the ocean model is higher near the equator (0.33°) and in the Southern Ocean (∼0.4° at 70°S). Ocean biogeochemistry is represented by the Whole Ocean Model of Biogeochemistry And Trophic-dynamics (WOMBAT, Oke et al. (2013)).

WOMBAT is a nutrient-phytoplankton-zooplankton-detritus (NPZD) model (Oke et al., 2013; Law et al., 2017; Ziehn et al., 2020). It includes one functional type of phytoplankton and zooplankton, dissolved inorganic carbon (DIC), alkalinity (ALK),

phosphate ($PO_4$), oxygen ($O_2$), and iron as prognostic tracers. The stoichiometry is fixed at a C:N:P:$O_2$ ratio of 106:16:1:-172. $CaCO_3$ export from the photic zone is set at ∼8% of the organic carbon export. Detrital decomposition is a function of temperature and is allowed to occur when oxygen is zero to simulate the effect of denitrification, even though nitrification and denitrification are not explicitly included in the model and the global nitrogen budget is kept constant (Oke et al., 2013). The dissolution of $CaCO_3$ occurs at a constant rate. All organic and inorganic particles reaching the bottom are remineralized, given

that ACCESS-ESM1.5 does not include burial of sediments.

The pre-industrial 1850 equilibrium simulation (PI) is integrated following the CMIP6 protocol (Eyring et al., 2016) using the CMIP5 solar constant (1365.65 W.m$^{-2}$). The MIS 5e simulation (LIG) is integrated following PMIP4 protocol (Otto-Bliesner et al., 2017) with the solar constant of CMIP5-PMIP3 (1365.65 W.m$^{-2}$). Both PI and LIG simulations have been



evaluated extensively in the recent literature (Ziehn et al., 2020; Otto-Bliesner et al., 2021; Kageyama et al., 2021; Yeung et al.,
2021; Mackallah et al., 2022; Choudhury et al., 2022; Yeung et al., 2024). The MIS 9e simulation was set up with boundary
conditions corresponding to 333 ka and the same solar constant as for PI and LIG (see Table 1). The Greenland and Antarctic
ice-sheets as well as vegetation distribution are the same in all three simulations, but leaf area index is calculated prognostically.
Anomalies in incoming solar radiation are shown in Figure A1.

Dissolved oxygen is a tracer with very long equilibration times. Figure A2 shows time series of dissolved oxygen for the
three simulations analysed in this study. While oxygen concentrations have reached quasi-equilibrium for North Atlantic Deep
Water (NADW), in the intermediate waters of the North Pacific, and in Antarctic Bottom Water south of 35°S in all runs, there
is still a slight drift in global mean dissolved oxygen in our LIG and MIS 9e simulations and a more substantial drift in the
Mediterranean Sea.

**Table 1.** Experimental set-up

| | PI | LIG | MIS 9e |
|---|---|---|---|
| Orbital parameters | | | |
| Eccentricity | 0.016764 | 0.039378 | 0.03229469 |
| Obliquity (degrees) | 23.459 | 24.040 | 24.2397 |
| Perihelion -180 | 100.33 | 275.41 | 297.994 |
| Vertical equinox | 21 March at noon | 21 March at noon | 21 March at noon |
| Greenhouse gases | | | |
| Carbon dioxide (ppm) | 284.3 | 275 | 298.6 |
| Methane (ppb) | 808.2 | 685 | 797 |
| Nitrous oxide (ppb) | 273 | 255 | 287.3 |
| Paleogeography | Modern | Modern | Modern |
| Ice sheets | Modern | Modern | Modern |
| Vegetation | PI | PI | PI |
| Aerosols | CMIP Deck piControl | CMIP Deck piControl | CMIP Deck piControl |
| Integration length | 1000 years | 1823 years[1] | 1750 years |
| Remaining drift below 3000m per 100 years | | | |
| Potential temperature (°C/ 100 years) | +0.004 | +0.040 | +0.055 |
| Dissolved $O_2$ (mmol.m$^{-3}$ / 100 years) | +0.38 | -0.85 | -1.28 |

[1] *The first 372 years have erroneous forcing*



## 3  Results

### 3.1  Large-scale oxygenation

All results reported in this section are based on 100-year means, calculated over the last 100 years of each simulation (see rectangles in Figure A2). The global temperature patterns simulated by ACCESS ESM1.5 under LIG boundary conditions are described elsewhere (Otto-Bliesner et al., 2021; Yeung et al., 2021, 2024). Figure A3 shows annual mean surface air temperature (SAT) anomalies and sea surface temperature (SST) anomalies for the simulations described in this study. Due to different orbital parameters, and in particular a positive summer insolation anomaly at high northern latitudes and a positive spring anomaly at high southern latitudes (Mitsui et al. (2022); Figure A1), the annual mean simulated surface air temperature in our LIG simulation is $1.11°C$ and $1.40°C$ higher north and south of $40°$, respectively, compared to PI . While the insolation during MIS 9e is similar to LIG (Mitsui et al., 2022), greenhouse gas concentrations are higher (Table 1), with anomalies of ~24 ppm for $CO_2$, ~112 ppb for $CH_4$ and ~32 ppb for $N_2O$, thus leading to globally warmer conditions in our MIS 9e simulation compared to the LIG simulation (Figure A3). The annual mean simulated surface air temperature is $0.49°C$ and $1.20°C$ higher in LIG and MIS 9e runs, respectively, compared to PI.

The PI simulation reproduces observed patterns of dissolved oxygen (World Ocean Atlas, 1965-2020) reasonably well (Ziehn et al. (2020); Figure 1c and d), but underestimates $O_2$ in the eastern Pacific Ocean and southeastern Atlantic Ocean. It overestimates $O_2$ in the Arabian Sea, the Southern Ocean and the Arctic. The global mean ocean $O_2$ concentration equals $176.3$ mmol·m$^{-3}$ in our PI simulation compared to $158.0$ mmol·m$^{-3}$ and $154.5$ mmol·m$^{-3}$ in the LIG and MIS 9e simulations, respectively. The largest differences between the simulations are located in the deep ocean (below 2500m depth) and follow the pathways of Antarctic Bottom Water (AABW; Figures 1, 2). Overall, the anomalies in LIG and MIS 9e compared to PI are quite similar, with the loss in AABW $O_2$ being slightly less pronounced in LIG (Figures 1, 2). Significant differences can also be seen in the Mediterranean Sea (Section 3.2).

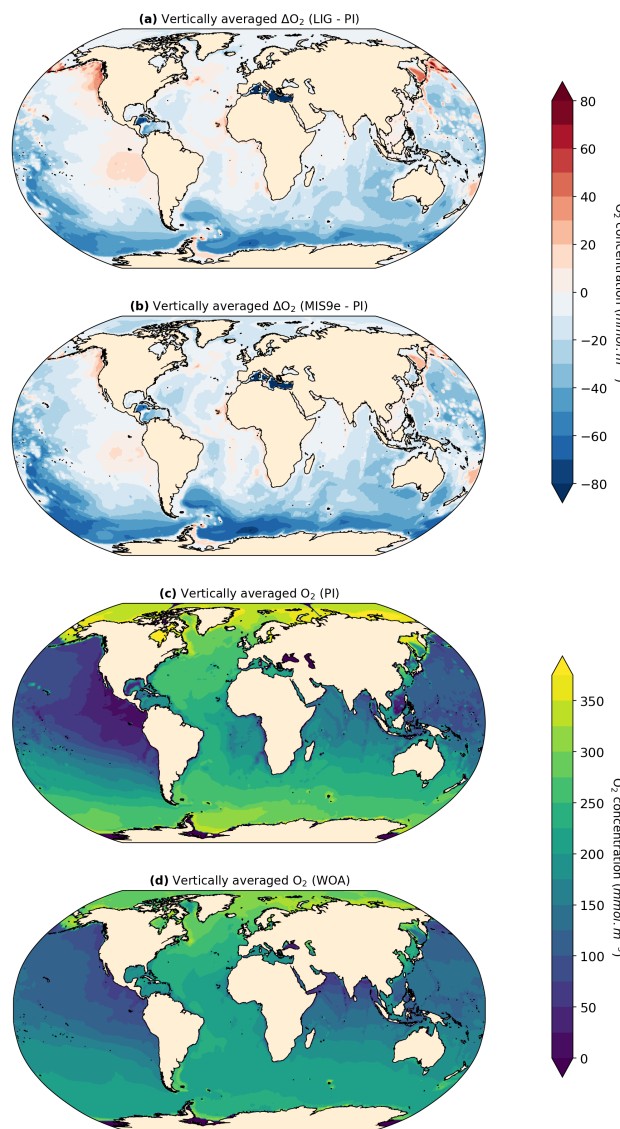

**Figure 1.** Vertically averaged dissolved $O_2$ concentration anomalies in mmol·m$^{-3}$. (a) LIG - PI, (b) MIS 9e - PI, (c) vertically averaged dissolved $O_2$ concentration for the PI simulation and (d) vertically averaged $O_2$ concentration above 5500m depth from World Ocean Atlas (WOA, 1965 - 2022, Reagan et al. (2025)).



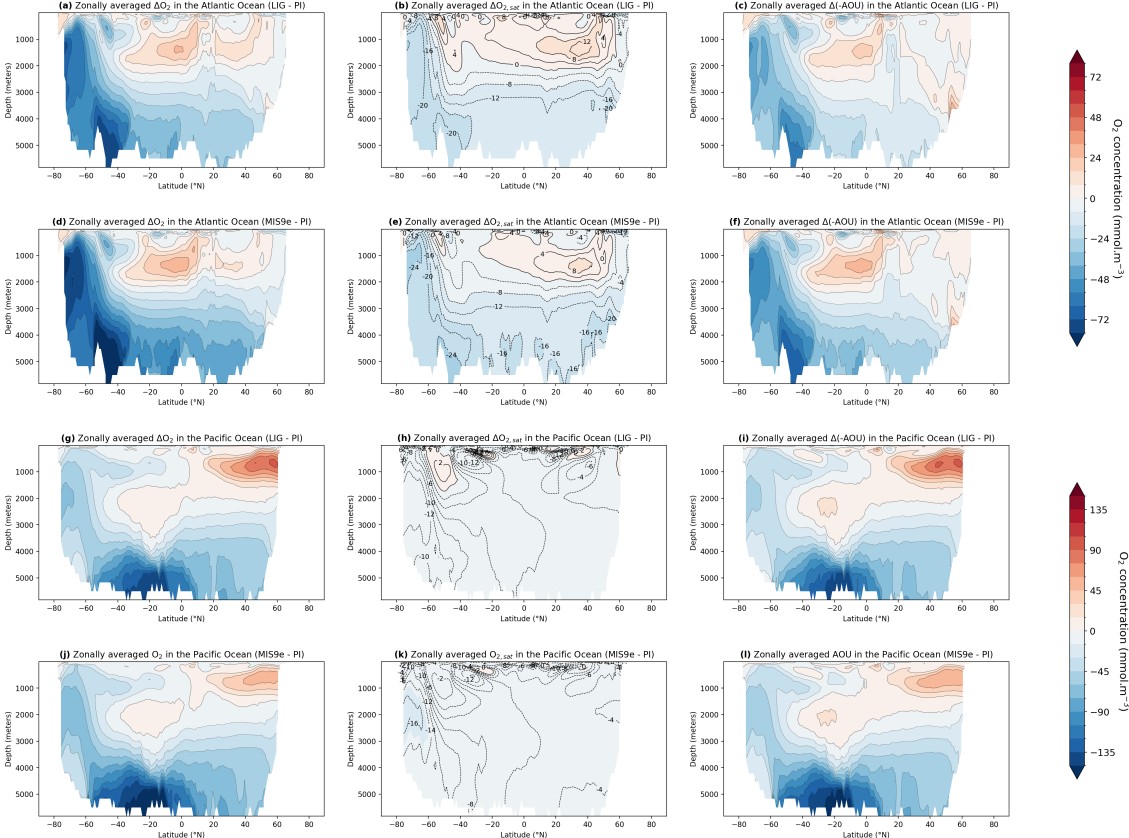

**Figure 2.** Zonally averaged dissolved $O_2$ concentration in mmol·m$^{-3}$ (left), saturated $O_2$ concentration (middle) and $-1$·AOU (Apparent Oxygen Utilisation, right). Atlantic Ocean anomalies LIG - PI (a-c), MIS 9e - PI (d-f); Pacific Ocean anomalies LIG - PI (g-i), MIS 9e - PI (j-l). To facilitate the comparison between oxygen solubility and oxygen utilization, we multiplied AOU by $-1$.

### 3.1.1 Antarctic Bottom Water

AABW is warmer and weaker in our LIG and MIS 9e simulations compared to PI (Figure A4). At 30°S, the maximum absolute value of the streamfunction reaches 3.8 Sv in LIG and 2.9 Sv in MIS 9e compared to 7.9 Sv in PI (Figure A5). Bottom water convection sites in the Weddell, Lazarev and Ross Seas are smaller in spatial extent and are also less intense (Figure A6). This weakening of deep-ocean convection is mainly due to reduced sea-ice formation (Yeung et al., 2024; Choudhury et al., 2022) and leads to warmer, less ventilated, and therefore less oxygenated AABW. The relative contribution of changes in Apparent Oxygen Utilisation (AOU) is higher than the contribution of changes in solubility (compare middle and right columns in Figure 2). Export production is enhanced in the Southern Ocean in LIG and MIS 9e compared to PI (Choudhury et al. (2022); Figure 3) and remineralisation rates are higher due to higher temperatures (Choudhury et al., 2022). Changes in dissolved oxygen concentrations in AABW are therefore primarily due to the weakening of Antarctic Bottom Water circulation, together





with the increase in export production and higher remineralisation rates, and secondarily to the temperature-depended solubility effect.

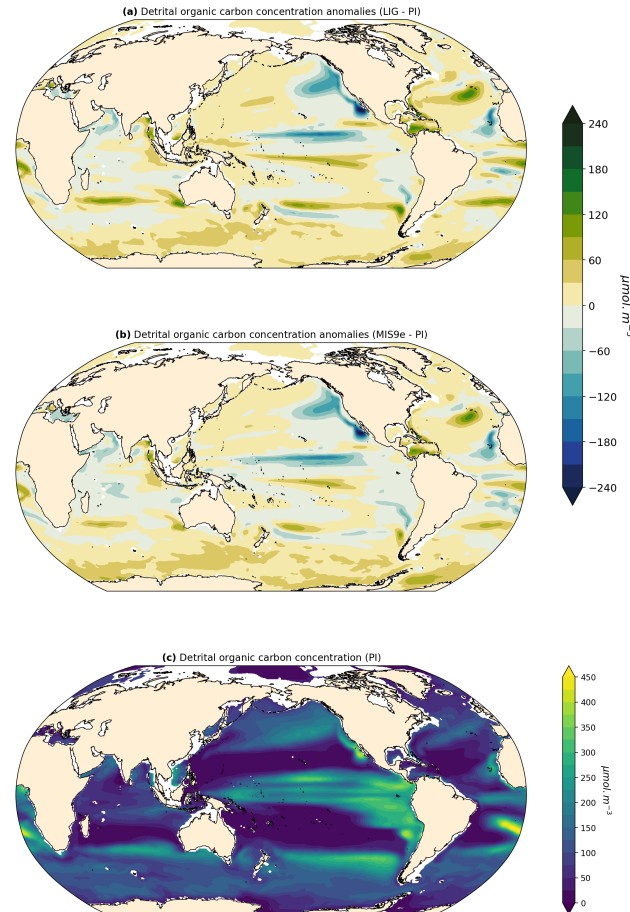

**Figure 3.** Detrital organic carbon concentration and anomalies in $\mu$molC·m$^{-3}$ at 250m depth. (a) LIG - PI, (b) MIS 9e - PI, and (c) PI.

### 3.1.2 North Atlantic Deep Water

Figure 2a and d shows that North Atlantic Deep Water (NADW) is more oxygenated in both MIS 9e and LIG compared to PI, with MIS 9e being more oxygenated than LIG at mid-depth between 40°S and 20°N. This is due to enhanced deepwater

formation (Choudhury et al. (2022); Figure A5) and stronger advection of NADW at depth leading to a decrease in AOU (Figure 2c and f), and to lower temperatures of NADW in LIG and MIS 9e compared to PI (Figure A4; Figure 2, middle column). The maximum value of the streamfunction in the Atlantic at 45°N reaches 23.6 Sv in our LIG simulation and 24.3 Sv in the MIS 9e simulation, compared to 19.9 Sv in the PI simulation. These simulated changes in NADW transport are associated with an advective-convective feedback mechanism involving the Subpolar Gyre (SPG), first described by Levermann and Born



(2007). The SPG is stronger in our MIS 9e and LIG simulations than in PI (Figure A7, left column), with an establishment of a third region of deep convection in the Labrador Sea in addition to the deep convection sites in the Norwegian Sea and south of the Greenland-Scotland Ridge (Figure A6). During boreal winter (December to February), a stronger SPG results in stronger outcropping of isopycnals and in higher sea surface salinities (Figure A7, right column) due to enhanced transport of subtropical saline waters, both are preconditioning the surface waters for convection. The resulting horizontal surface density

gradients maintain the strong gyre. The new convection site in the Labrador Sea contributes colder waters to the resulting North Atlantic Deep Water mass, leading to an overall colder NADW compared to PI (Figure A4). Export production is enhanced over most of the North Atlantic Ocean in LIG and MIS 9e compared to PI and is therefore not one of the drivers of the increase in dissolved oxygen (Figure 3).

### 3.1.3 North Pacific Intermediate Water

The North Pacific ocean above ~2000m is significantly more oxygenated in our LIG and MIS 9e simulations than in our PI simulation (Figure 2g and j), with LIG being even more oxygenated than MIS 9e. The potential temperatures of the affected water masses are higher in both simulations compared to PI (Figure A4; Figure 2h and k). The oxygenation is therefore solely due to changes in AOU which shows strong negative anomalies (Figure 2i and l). We can see a slight increase in ventilation during winter months in both simulations due to higher subduction rates in the western Bering Sea - along the coast of the

Kamchatka peninsula (Figure A8). This leads to younger and better ventilated waters at 250m depth circulating across the Pacific and then southward following the California Current (shown in Figure A9). There is also a significant decrease in export production off the west coast of North America in LIG and MIS 9e compared to PI (Figure 3). This is due to a weakening of the north-westerly winds causing a weakening of the coastal upwelling zones (Figure A10, left column) contributing significantly to the gain in oxygen at deeper layers (Figure 1a and b).

### 3.1.4 Oxygen Minimum Zones

Figure 4 (left column) shows dissolved $O_2$ concentrations in hypoxic zones at 300m depth. In our PI simulation, the model represents the large-scale patterns reasonably well in the tropical and subtropical Pacific, but overestimates oxygen depletion within these zones (Figure 4e and g). The simulated OMZ in the Atlantic ocean off the African coast is also overestimated in extent and intensity. Contrarily, oxygen depletion in the North Pacific and the Arabian Sea is underestimated at 300m depth,

but the North Pacific ocean becomes hypoxic at deeper levels (Figure 4, right column).






**Figure 4.** $O_2$ concentration and anomalies at 300m depth in hypoxic zones[1] (left) and vertical minimum of $O_2$ concentration and anomalies in hypoxic zones[2] (right) in (a, b) LIG - PI, (c, d) MIS 9e - PI, (e, f) PI and (g, h) World Ocean Atlas (WOA, 1965-2022). Black contour lines in subplots a-d indicate the 62 mmol·m$^{-3}$ isolines for PI (solid), LIG (dotted), and MIS 9e (dashed).

[1]*hypoxic zones defined as zones where $O_2$ concentration at 300m is below 62 mmol·m$^{-3}$.*

[2]*hypoxic zones defined as zones where vertical minimum of $O_2$ concentration is below 62 mmol·m$^{-3}$.*



The extent and intensity of OMZs is very similar between the three simulations at 300m depth (Figure 4a and c). The Eastern Boundary Upwelling System off the coast of North America is weakened in our LIG and MIS 9e simulations compared to PI (Figure A10), which leads to a decrease in nutrient availability at the surface (not shown), primary productivity, and export production (Figure 3). Consequently, AOU is significantly reduced in our LIG and MIS 9e simulations off the coast

of California and Baja California (Figure A11b and d) leading to an increase in oxygen in the northern hemispheric tropical Pacific OMZ that is partially compensated by a decrease in solubility due to higher water temperatures (Figure A11a and c). The Eastern Boundary Upwelling System off Africa is also weakened (Figure A10, right column), but does not affect the OMZ, which is situated further south. The OMZs in the southern hemispheric tropical Pacific and South Atlantic are slightly intensified in our LIG and MIS 9e simulations compared to PI, which is due to warmer waters and partially compensated by a

decrease in AOU in these regions (Figure A11).

The right column of Figure 4 shows the minimum of dissolved oxygen in the water column. The model represents the extent of hypoxic waters well in the Pacific Ocean but underestimates the extent in the Indian Ocean, and, again, misses the OMZ in the Arabian Sea. The extent of the South Atlantic OMZ is overestimated. The intensity of oxygen loss is overestimated in the eastern regions of all OMZs (Figure 4f and h). Differences between the simulations show that the mid to high latitude North

Pacific is significantly more oxygenated in LIG and, to a lesser extent, in MIS 9e, as discussed in Section 3.1.3. The southern and western parts of the Pacific hypoxic zones as well as most of the hypoxic zones in the Atlantic Ocean become more oxygen depleted in LIG and MIS 9e compared to PI. The minimum oxygen concentration in the water column shifts to deeper depths in the northern hemispheric Pacific and to shallower depth in most of the southern hemispheric Pacific (Figure A12).

**Table 2.** Total volume (percentage) of oxygen depleted water masses.

|  | PI | LIG | MIS 9e |
| --- | --- | --- | --- |
| Hypoxic ($<62$ mmol·m$^{-3}$) | $1.95 \cdot 10^8$ km$^3$ (15.13%) | $2.50 \cdot 10^8$ km$^3$ (19.40%) | $2.76 \cdot 10^8$ km$^3$ (21.43%) |
| Suboxic ($<10$ mmol·m$^{-3}$) | $6.70 \cdot 10^7$ km$^3$ (5.20%) | $4.61 \cdot 10^7$ km$^3$ (3.58%) | $5.99 \cdot 10^7$ km$^3$ (4.65%) |
| Anoxic ($<1$ mmol·m$^{-3}$) | $3.99 \cdot 10^7$ km$^3$ (3.10%) | $2.77 \cdot 10^7$ km$^3$ (2.15%) | $2.96 \cdot 10^7$ km$^3$ (2.30%) |

The global ocean is overall less oxygenated in MIS 9e and LIG compared to PI and this is also reflected in the total volumes

of hypoxic waters ($<62$ mmol·m$^{-3}$), which is 5% larger in these simulations (Table 2). However, the suboxic ($<10$ mmol·m$^{-3}$) and anoxic zones ($<1$ mmol·m$^{-3}$) are smaller in our LIG and MIS 9e simulations, mainly due to the oxygenation of the North Pacific Ocean (Section 3.1.3).

### 3.2 Oxygenation of the Mediterranean Sea

The mean O$_2$ concentration in the Mediterranean Sea equals 218.7 mmol·m$^{-3}$ in our PI simulation compared to 106.2

mmol·m$^{-3}$ and 61.6 mmol·m$^{-3}$ in our MIS 9e and LIG simulations, respectively. 63% of Mediterranean waters are hypoxic, with dissolved oxygen concentrations below 62 mmol·m$^{-3}$, and 27% are anoxic (less than 1 mmol·m$^{-3}$) in the LIG simula-

 

tion. The MIS 9e simulation is not quite as depleted in oxygen in the Mediterranean Sea, with 34% of the total volume being hypoxic and 5.6% anoxic (Table 3).

**Table 3.** Volume (percentage) of oxygen depleted water masses in the Mediterranean Sea

|  | PI | LIG | MIS 9e |
|---|---|---|---|
| Hypoxic ($<62$ mmol·m$^{-3}$) | 0 | $1.88\cdot10^6$ km$^3$ (62.63%) | $1.01\cdot10^6$ km$^3$ (33.65%) |
| Suboxic ($<10$ mmol·m$^{-3}$) | 0 | $1.14\cdot10^6$ km$^3$ (38.05%) | $1.90\cdot10^5$ km$^3$ (6.34%) |
| Anoxic ($<1$ mmol·m$^{-3}$) | 0 | $8.14\cdot10^5$ km$^3$ (27.08%) | $1.67\cdot10^5$ km$^3$ (5.55%) |

The depletion of dissolved oxygen in the MIS 9e and LIG simulations is greatest in intermediate water masses (Figures 5c, 6, right column), with horizontally averaged O$_2$ concentrations dropping to 50 mmol·m$^{-3}$ in MIS 9e and reaching anoxic values in LIG between 500m and 1000m depth (Figure 5c). This deoxygenation is caused by a strong pycnocline due to a significant freshening and warming of the surface layers (Figure 5a and b) that prevents mixing between surface and deeper layers and increases the water ages of intermediate and deep water masses (Figure 5d). While there is deep water formation in winter months in the Ionian Sea with seasonal mean winter ventilation depth reaching 489 m, and to a lesser extent in the southern Adriatic Sea and east of the Gulf of Lion (west of Corsica) in our PI simulation, winter mean mixed layer depths do not exceed 97 m and 106 m in our LIG and MIS 9e simulations, respectively (Figure A13).

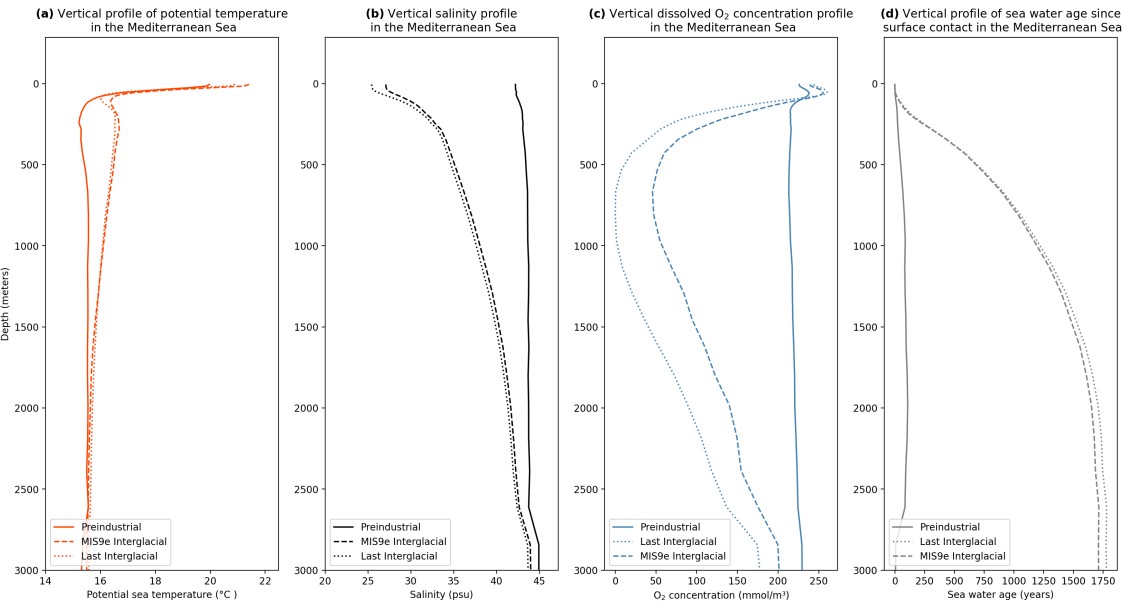

**Figure 5.** Zonal and meridional mean vertical profiles in the Mediterranean Sea for PI, LIG and MIS 9e. (a) Potential temperature in °C, (b) salinity in psu, (c) O$_2$ concentration in mmol·m$^{-3}$ and (d) seawater age since last surface contact in years. Note that the maximum possible water age equals the length of the simulation and might therefore underestimate the water age for very old water masses.




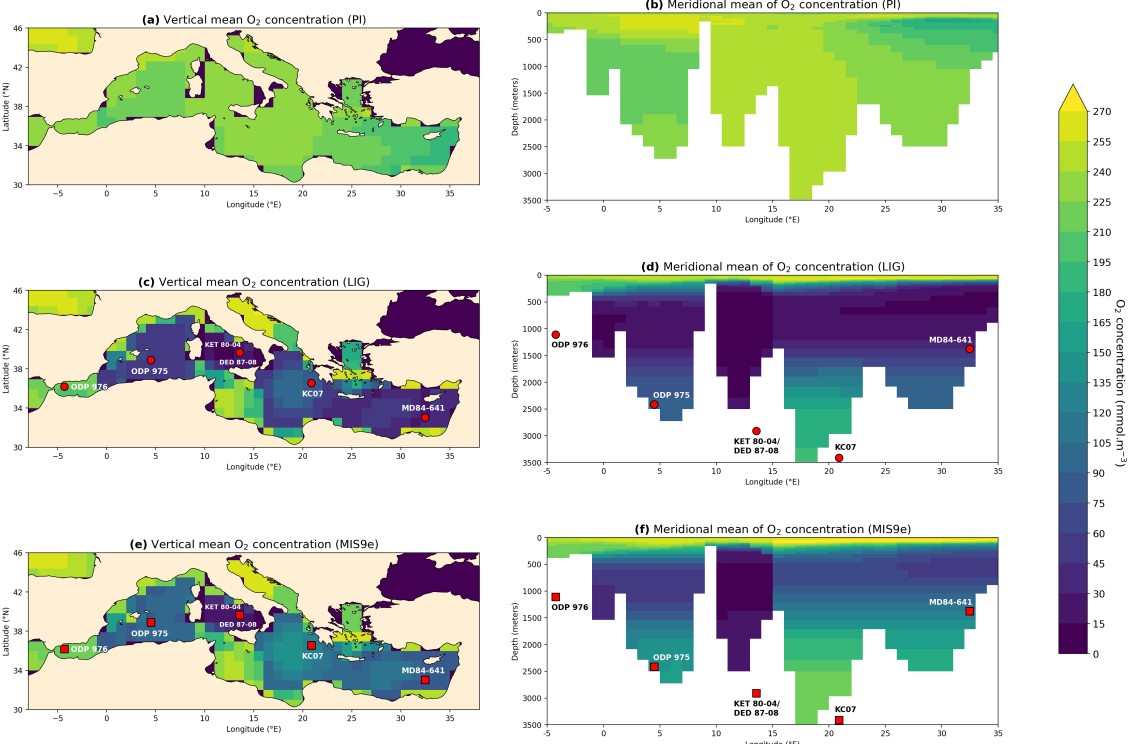

**Figure 6.** Vertical mean O$_2$ concentration (left) and meridional mean O$_2$ concentration (right) in the Mediterranean Sea in mmol·m$^{-3}$ for PI (a, b), LIG (c, d) and MIS 9e (e, f). Symbols represent core sites discussed in the text.

Figure 6 shows that the central Mediterranean Sea is severely oxygen depleted in both LIG and MIS 9e simulations with O$_2$ concentrations dropping to zero in the Tyrrhenian Sea below 100m depth. This is in qualitative agreement with the appearance of organic rich dark layers in core DED 87-08 (39°42'N, 13°35'E, 2,965m, Fontugne and Calvert (1992); Kallel et al. (2000))
during eastern Mediterranean Sapropel events S5 (121.5-128.3 ka BP, Grant et al. (2016)) and S10 (∼332 ka BP, Konijnendijk et al. (2014)). Core KET 80-04 (39°40'N, 13°34'E, 2,909m) also shows evidence of anoxic conditions during S5 in the Tyrrhenian Sea (Kallel et al., 2000).

The Eastern Mediterranean basin is also oxygen depleted, especially in our LIG simulation. The simulated oxygen depletion in the Levantine Sea is in agreement with core MD 84-641 (33°02'N, 32°38'E, 1,375m) which recorded a strong sapropel event
(S5) at the LIG (4.52% total organic carbon (TOC)) and a less strong sapropel event (S10) at MIS 9e (2.5% TOC) (Fontugne and Calvert, 1992; Melki et al., 2010). Our results are also in agreement with Andersen et al. (2018), who find water mass ages of 1030 (+820/-520 years) in the deep Eastern Mediterranean Sea (compared to ∼ 100 years today) at the end of S5, and with Sweere et al. (2021) who find euxinic conditions during S5 in the Levantine Sea.

The Ionian Sea is better oxygenated than the Levantine Sea in our simulations, but conditions at intermediate depths are still
hypoxic in the MIS 9e simulation dropping to 52.4 mmol·m$^{-3}$ and reaching anoxic values in the LIG simulation. Core KC07



(20°53'N, 36°34'E, 3,410m) records rich organic layers during both S5 and S10 (Köng et al., 2017) but is situated at a depth where our simulations show higher oxygen concentrations. It should be noted that oxygen concentrations in the deep waters in the Ionian Sea are still drifting in our simulations and might reach suboxic values if integrated for longer.

In the western Mediterranean, sapropels are less distinctive than in the eastern Mediterranean, but they can still be detected in terms of higher-than-background organic carbon content. ODP leg 161 drilled five sites throughout the western Mediterranean and found that sapropel TOC values decrease to the west and are lowest in the Alboran Sea (Murat, 1999). Cores 975 (38°53.795'N, 04°30.595'E, 2,416m), 976 (36°12.32'N, 04°18.760'W, 1,108m), 977 (36°01.907N, 01°957.319'W, 1,984m) and 979 (35°43.427'N, 03°12.353'W, 1,062m) all record higher organic carbon layers between 122 and 128ka BP, and between 328 and 331ka BP, and the S5 event is colour-banded in site 975, while the S10 event is homogeneous (Murat, 1999). This east-west gradient is reflected in our simulations, where waters near the Strait of Gibraltar are better oxygenated than waters in the Alboran Sea.

The North African monsoon expands significantly into the Sahara region in our LIG (Yeung et al., 2021) and MIS 9e simulations, hence causing an intensification of precipitation during the wet season in North Africa (Yeung et al., 2021; Menviel et al., 2021). Rainfall in the Sahara region increases by up to +14.4 mm·d$^{-1}$ in LIG and +12.7 mm·d$^{-1}$ in MIS 9e during boreal summer (JJA) compared to PI (Figure 7a and b). This excess rainfall causes excess freshwater runoff that enters the Mediterranean basin mainly through the Nile delta. Although North African monsoon intensity peaks between June and August, outflow into the Mediterranean Sea reaches its maximum in autumn between September and November (Figure 7c and d). This time lag of about 3 months is caused by the river routing timescale embedded in the ACCESS-ESM1.5 model (Total Runoff Integrating Pathways (TRIP) river routing, Oki et al. (1999)) and is similar to the observed timescale of river routing.





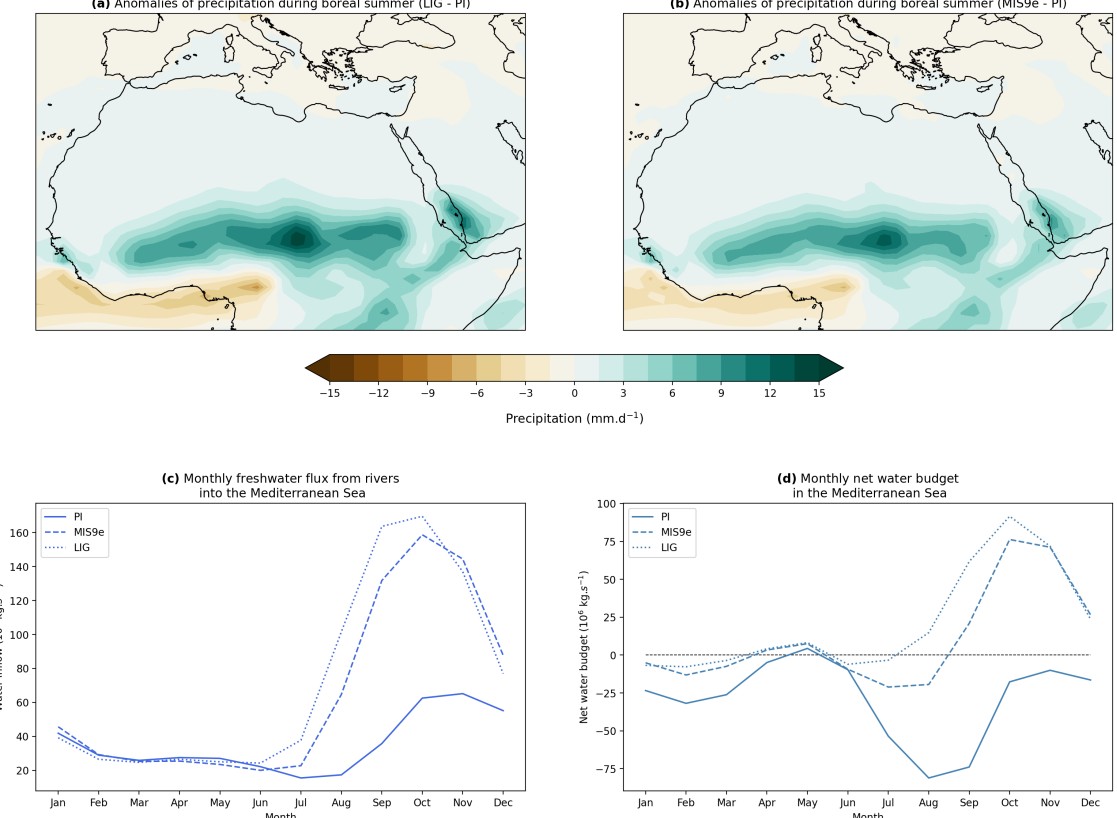

**Figure 7.** Anomalies of precipitation in Northern Africa during boreal summer (JJA) in mm·d$^{-1}$ for (a) LIG - PI and (b) MIS 9e - PI; (c) seasonal freshwater discharge from all rivers into the Mediterranean Sea for PI, LIG and MIS 9e in $10^6$ kg·s$^{-1}$ and (d) seasonal freshwater surface fluxes (precipitation + runoff - evaporation) into the Mediterranean Sea in $10^6$ kg·s$^{-1}$ for PI, LIG and MIS 9e.

The annual mean surface freshwater flux into the Mediterranean Sea (precipitation + river run-off - evaporation) equals -344·$10^6$ kg·s$^{-1}$ in our PI simulation compared to +248·$10^6$ kg·s$^{-1}$ and +130·$10^6$ kg·s$^{-1}$ in our LIG and MIS 9e simulations, respectively. These positive fluxes cause freshening of the surface layers, stratification and, eventually, deoxygenation of the Mediterranean Sea in our simulations (Figure A2b). Compared to MIS 9e, the North African monsoon intensification is more pronounced in the LIG simulation (Figure 7a and b), leading to higher river runoff (Figure 7c). Sea surface salinities decrease at a faster rate and lead to stratification and deoxygenation earlier than in the MIS 9e simulation (Figure A2b). It should be noted that oxygen levels in the Mediterranean Sea are still drifting in both simulations and that the equilibrium values are lower than what is presented here.





## 4   Discussion

The oxygenation of the world's oceans is very similar in our LIG and MIS 9e simulations and quite different compared to our
PI simulation. These differences are primarily due to circulation changes and changes in export production and secondarily
to the solubility effect. The large-scale ocean circulation patterns, including AABW, NADW and the ventilation of the North
Pacific Ocean are therefore very sensitive to the latitudinal and seasonal distribution of incoming solar radiation in the ACCESS
ESM1.5, and less sensitive to greenhouse gas concentrations. The insolation anomalies are indeed quite similar for LIG and
MIS 9e (Figure A1), while the greenhouse gas forcing is highest for MIS 9e, lowest for LIG, with PI being in the middle
(Table 1).

While quantitative records of past oxygenation are difficult to reconstruct with proxies (Hoogakker et al., 2024), qualita-
tive records exist. For the more recent Pleistocene period, most records looking at past ocean oxygenation have focused on
glacial-interglacial variations; for summaries see Jaccard and Galbraith (2012) and Moffitt et al. (2015). Stott et al. (2000) used
epibenthic foraminifera carbon isotopes to make inferences of intermediate water ventilation and oxygenation in the northeast
Pacific, and suggested that waters above 2000m depth were less oxygenated during MIS 5e. If the proposed relationship be-
tween benthic foraminifera carbon isotopes and apparent oxygen utilization holds, then overall reduced carbon isotopes during
MIS 5e (Bengtson et al., 2021) would imply an overall decrease in global ocean oxygenation, agreeing with our simulations.
There is also evidence for reduced AABW formation and a decrease in dissolved oxygen in the Southern Ocean based on redox
elements (Hayes et al., 2014; Glasscock et al., 2020).

Oxygen Minimum Zones (OMZs) have expanded and contracted repeatedly in the past in response to global changes in
climate (Jaccard and Galbraith, 2012). To the authors' knowledge, there is no global modelling study analysing the extent and
magnitude of OMZs during MIS 5e and MIS 9e. Simulating OMZs with Earth System Models is very challenging, because
low oxygen concentrations are the result of a subtle balance between two large opposing processes: the physical transport of
oxygen-rich waters into the region of interest and the local flux of particular matter and remineralization. Climate models are
not very skilled in representing these dynamics, mostly because their resolution does not resolve equatorial dynamics well
enough. Another problem is the simple representation of biology and geochemistry in the majority of current state-of-the-art
models. The ACCESS ESM1.5 is not an exception. While the extent of the OMZs in the tropical Pacific is reasonably well
represented, there are deficits in other regions. Our results show that relatively small changes in circulation, such as the subtle
increase in ventilation of the upper North Pacific Ocean in our LIG and MIS 9e simulations, can have a significant impact
on oxygen concentrations, emphasizing the need of a realistic representation of physical circulation when simulating oxygen.
For example, cores LV-28-44-3 (52°02.514'N, 153°05.949'E; 684 m) in the Eastern Sea of Okhotsk (Matul et al., 2016) and
MR0604-PC07A (51°16'56N, 149°12'60E, 1,247 m) in the central Sea of Okhotsk (Jimenez-Espejo et al., 2018) recorded
intervals of suboxic conditions during MIS 5e. While our simulation points to suboxic waters east of the Kamchatka peninsula,
the waters west of the peninsula are well oxygenated (Figure 4). It should be noted that these cores were retrieved from a semi-
isolated marginal basin and are influenced by small-scale local circulation patterns that are not well resolved in the ACCESS
ESM1.5.



We find that the main Eastern Boundary Upwelling Systems in the North Pacific and Atlantic Oceans are weaker in the LIG and MIS 9e simulations compared to PI. This is consistent with high sea surface temperatures and faunal composition off the Canary islands during the LIG (Muhs et al., 2014; Maréchal et al., 2020) and microfossil assemblages from MIS 5e off the coast of northern California (Poore et al., 2000). However, Si/Ti, Cd/Al and Ni/Al records from core MD02-2508 (23°27.91' N, 111°35.74' W, 606 m) suggest that biological productivity was high during MIS 5e further south, off the coast of Baja California (Cartapanis et al., 2014), which is contrary to our results, although some upwelling is still present in our LIG and MIS 9e simulations. Coastal upwelling may have been similar or enhanced during MIS 5e and MIS 9e along the southeast African Margin (Pichevin et al., 2005; Ufkes and Kroon, 2012), expanded along the Brazil margin during MIS 5e (Lessa et al., 2017), and increased at higher latitudes during MIS 5e and MIS 9e (Yao et al., 2024). This agrees broadly with the simulated patterns of detrital organic carbon concentration anomalies.

The Mediterranean's sedimentary record is punctuated by periodic deep-sea anoxic events (Sachs and Repeta, 1999; Rohling et al., 2015; Rush et al., 2019) that manifest themselves in layers with elevated organic carbon concentrations (sapropels) and are strongly associated with times of African monsoon intensification. We find monsoon intensification in our LIG and MIS 9e simulations, enhanced river run-off, and a freshening of the surface layers of the Mediterranean Sea. The resulting vertical stratification leads to anoxic conditions in the east Mediterranean Sea in our LIG simulation, and hypoxic conditions in our MIS 9e simulation. The Tyrrhenian Sea is anoxic in both simulations and the west Mediterranean shows an east-west gradient of oxygenation that is qualitatively in agreement with sediment data. It should be noted, however, that the ACCESS ESM1.5 is a global model with relatively coarse horizontal resolution (1°x1° in the Mediterranean Sea). Such coarse-resolution model cannot be expected to simulate Mediterranean circulation skilfully. There are three main sites of deep water formation in the Mediterranean Sea in today's climate. Western Mediterranean deep water is formed during winter in the Gulf of Lion. Eastern Mediterranean deep water formation occurs in two sites, the Adriatic Sea, where bottom water is formed in winter and exits through the Otranto strait, and the Rhodes gyre in the Levantine Basin. In our PI simulation, the main deep water formation occurs in the Ionian Sea, and to a lesser extent in the southern Adriatic Sea and east of the Gulf of Lion. Our results should therefore be taken as a qualitative indication of enhanced stratification and oxygen loss without putting too much trust into simulated regional changes. For example, Tesi et al. (2024) find shallow water euxinia during S5 on the Adriatic Shelf due to a shut down of North Adriatic Dense Water formation. This regions remains well ventilated in our simulations. In addition, note that Mediterranean oxygen content is still drifting in our LIG and MIS 9e runs. As a result, MIS 9e could reach similar oxygen loss as LIG on a longer timescale (Figure A2).

Here we perform a snapshot experiment of the peak LIG and MIS 9e climates with a single model. These simulations thus only represent an equilibrium response to peak interglacial conditions, and do not represent potential transient climatic changes associated with the end of deglaciations, or abrupt climate changes linked to continental ice-sheet melting. In addition, the dissolved $O_2$ content simulated in these experiments is very dependent of the simulated oceanic circulation. The state of the ocean circulation during these interglacials is, however, unclear. Transient simulations through a whole interglacial and multi-model intercomparisons would be preferable and give a better indication of the robustness and uncertainties of past oxygenation patterns and their variability (Cartapanis et al., 2014). We hope that other modelling groups will consider to



analyse ocean oxygenation during past interglacials. Oxygen concentration is a variable with extremely long equilibrium times, so caution should be taken when climate models are not integrated for at least 1,000 years, preferably longer, and the drift in the deep ocean is still significant.

## 5 Conclusions

We integrated three equilibrium simulations with Australia's Earth System Model ACCESS ESM1.5 under pre-industrial (PI), Last Interglacial (LIG, 127 ka BP) and MIS 9e (333 ka BP) boundary conditions. The LIG and MIS 9e simulations show similar anomalies in large-scale oxygenation pointing to the fact that circulation patterns and oxygen concentrations are more sensitive to the distribution of incoming solar radiation than to greenhouse gas concentrations. Antarctic Bottom Water (AABW) is weaker and warmer, leading to deoxygenation mostly due to higher oxygen utilization, and, to a lesser extent, the solubility effect. North Atlantic Deep Water (NADW) is overall cooler and better ventilated, leading to higher in situ oxygen concentrations. Water masses in the upper 2000m of the North Pacific are both warmer and higher in oxygen content, due to enhanced subduction in the Bering Sea and a reduction in export production off the coast of North America. These anomalies are more pronounced in the MIS 9e simulation than in the LIG simulation, with the exception of the North Pacific Ocean, where the oxygenation anomaly is strongest in the LIG simulation. While the global ocean is overall less oxygenated in MIS 9e and LIG compared to PI, the volume of suboxic and anoxic waters is smaller in MIS 9e and LIG, mainly due to the oxygenation of the North Pacific Ocean. The Mediterranean Sea is the exception, where oxygen is significantly more depleted in MIS 9e and LIG compared to PI, due to an intensification and expansion of the African Monsoon, enhanced run-off and resulting freshening of surface waters and stratification. In the LIG simulation, 30% of the total seawater volume in the Mediterranean Sea is anoxic ($< 1$ mm·m$^{-3}$) and 63% hypoxic ($< 62$ mm·m$^{-3}$). This deoxygenation is not quite as pronounced in the MIS 9e simulation with 6% of anoxic waters and 34% of hypoxic waters. Dissolved oxygen concentrations are still drifting in both simulations, even after integration times of well over 1,500 years.

The simulation of ocean oxygen concentrations is challenging, as oxygen levels depend on complex physical and biogeochemical processes that must be represented correctly in a climate model to simulate oxygen realistically. State-of-the-art Earth System Climate Models systematically underestimate the observed rates of oxygen loss and are also not very skillful in reproducing the observed patterns of deoxygenation (Oschlies et al., 2017, 2018). In this study, we confirmed that subtle changes in ocean circulation can have significant impact on oxygen concentrations. For trustworthy future projections, high-resolution ocean models are therefore needed to represent the relevant ocean circulation patterns as well as possible, but these models are computationally expensive. Oceanic oxygen takes centuries to adjust, and we currently lack computer power to run high-resolution models for the timescales needed for initialisation and equilibrium responses. Another important challenge is the representation of biogeochemical processes in models, as our knowledge of coupled biogeochemical processes is still incomplete.



# Appendix A

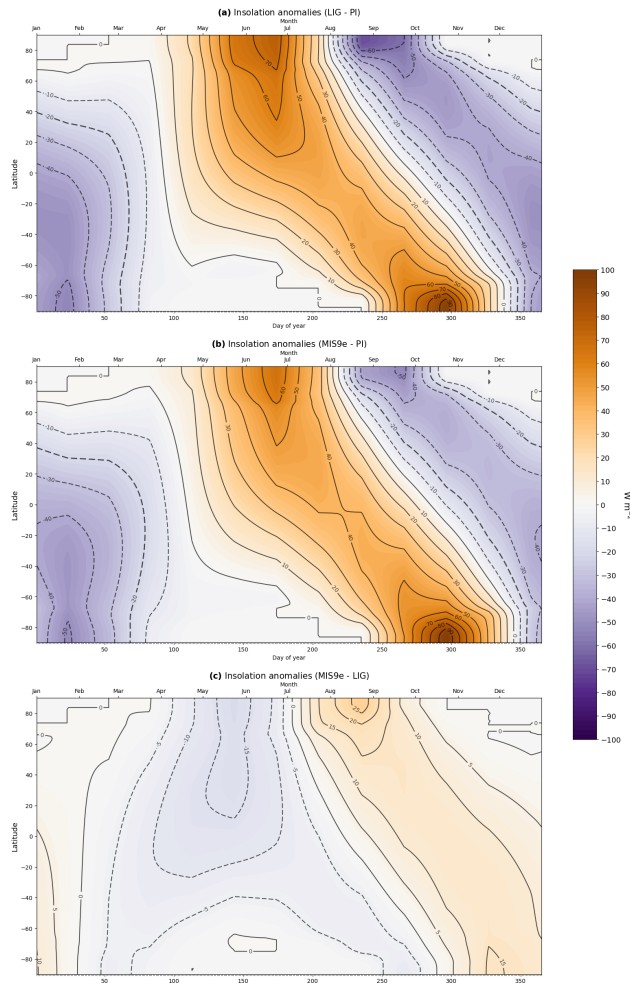

**Figure A1.** Insolation anomalies (W·m$^{-2}$) across latitudes and days of year. (a) LIG minus PI, (b) MIS 9e minus PI, (c) MIS 9e minus LIG. The bottom x-axis represents the day of year and the top x-axis represents months according to present-day calendar.




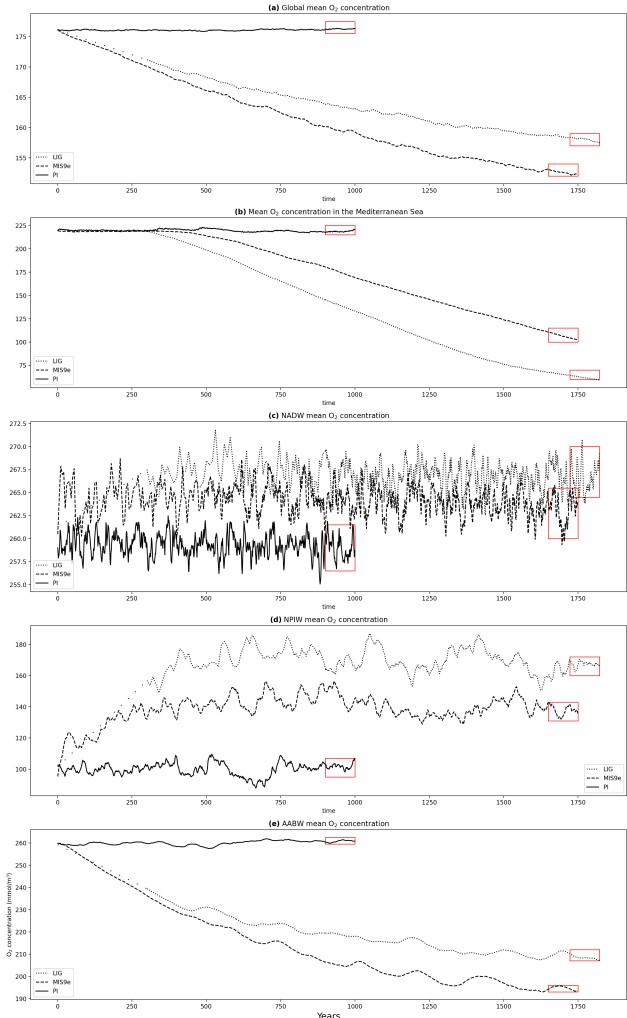

**Figure A2.** Timeseries of dissolved O$_2$ concentration in mmol·m$^{-3}$ averaged (a) globally, (b) in the Mediterranean Sea, (c) between 500m and 1200m depth, and 40°N-60°N (North Atlantic Deep Water), (d) between 250m and 1200m depth, and 40°N-60°N in the North Pacific, and (e) at 3000m depth, between 60°S and 35°S (Antarctic Bottom Water) for PI, LIG and MIS 9e simulations. Red rectangles designate the last 100 years of each simulation that are analysed in this manuscript. The first 372 years of LIG were integrated with erroneous boundary conditions and have been deleted from our file servers; we show therefore interpolations over this time span.





**Figure A3.** Near-surface air temperature (SAT) and anomalies (left) and sea surface temperature (SST) and anomalies (right) in °C. (a, b) LIG - PI, (c, d) MIS 9e - PI, (e, f) MIS 9e - LIG, (g, h) PI.





**Figure A4.** Zonally averaged potential temperature in °C (g, h) and anomalies (a-f) in the Atlantic Ocean (left) and in the Pacific Ocean (right). (a, b) LIG - PI, (c, d) MIS 9e minus PI, (e, f) MIS 9e minus LIG, (g, h) PI.





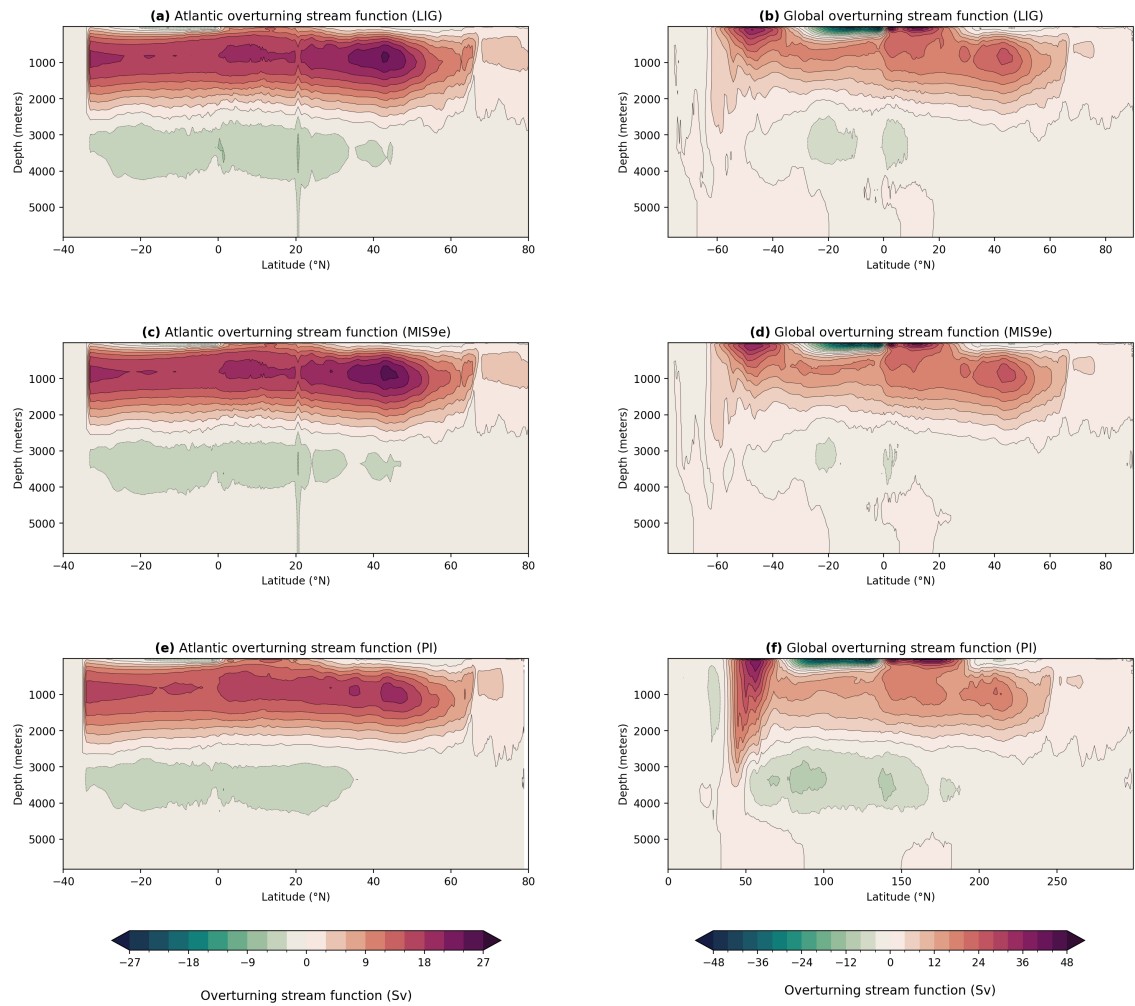

**Figure A5.** Meridional overturning streamfunction in Sv for the Atlantic Ocean (left) and global (right). (a, b) LIG, (c, d) MIS 9e , (e, f) PI.



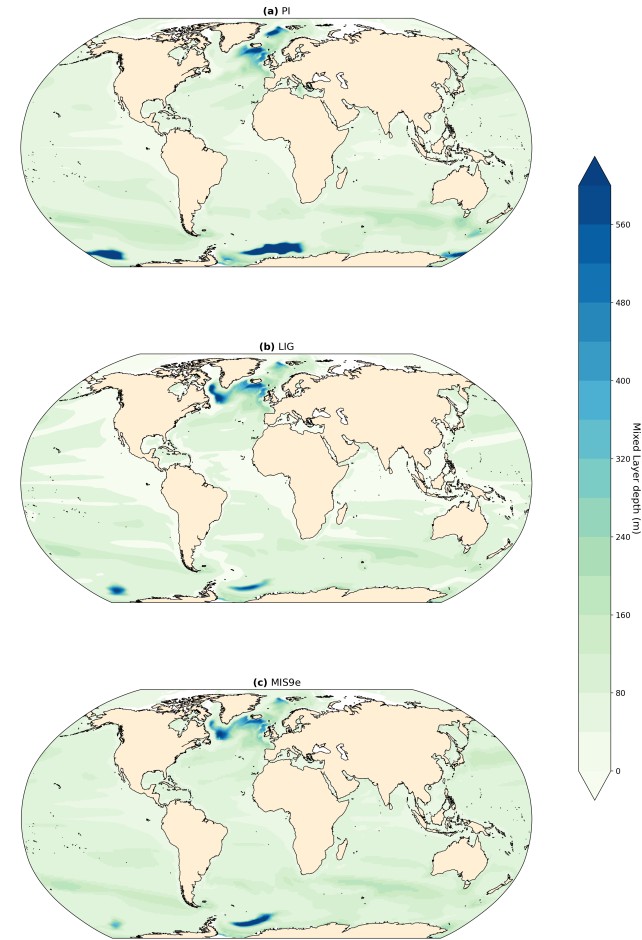

**Figure A6.** Annual mean mixed layer depth in m. (a) PI, (b) LIG and (c) MIS 9e.




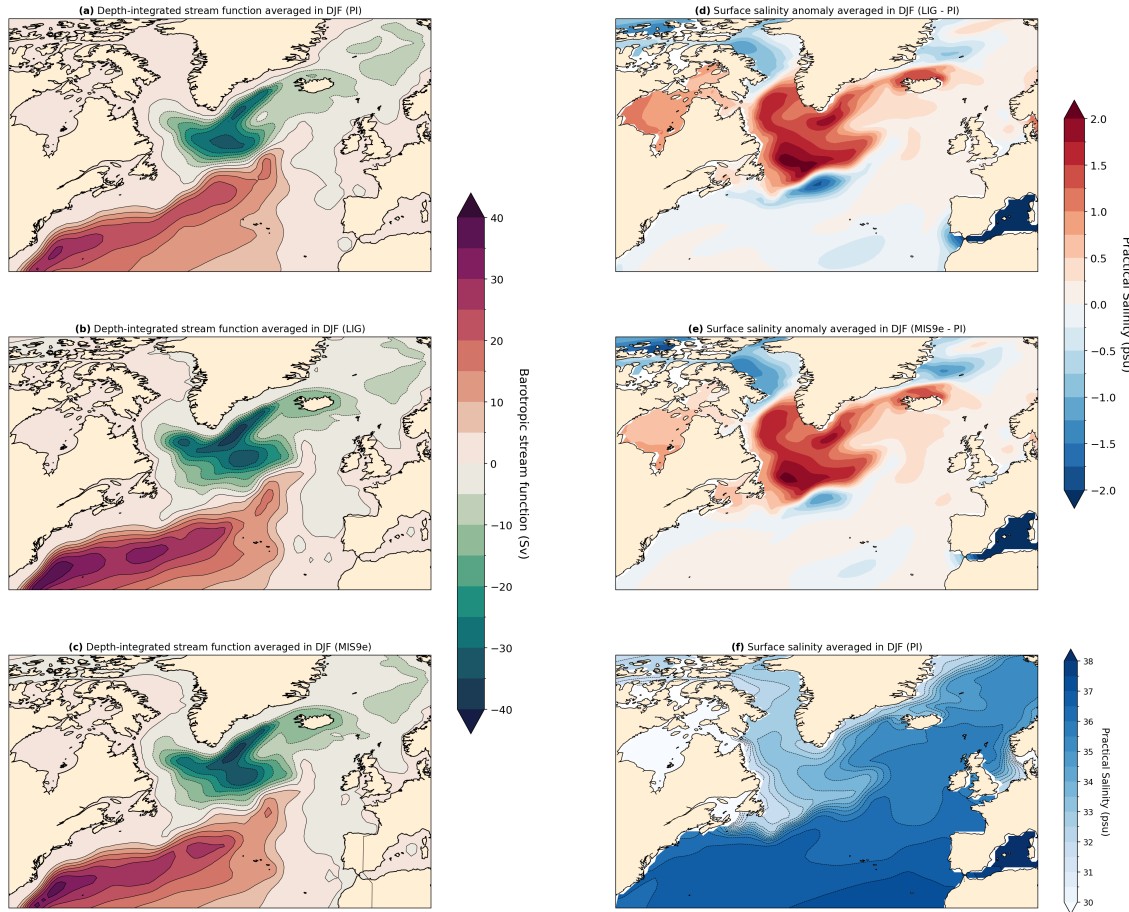

**Figure A7.** Depth-integrated barotropic stream function in the North Atlantic Ocean averaged during boreal winter in Sv (DJF, a-c) and sea surface salinity in psu (f) and anomalies (d, e) in the North Atlantic Ocean averaged over boreal winter months (DJF). (a) PI, (b) LIG, (c) MIS 9e, (d) LIG - PI, (e) MIS 9e - PI, (f) PI.





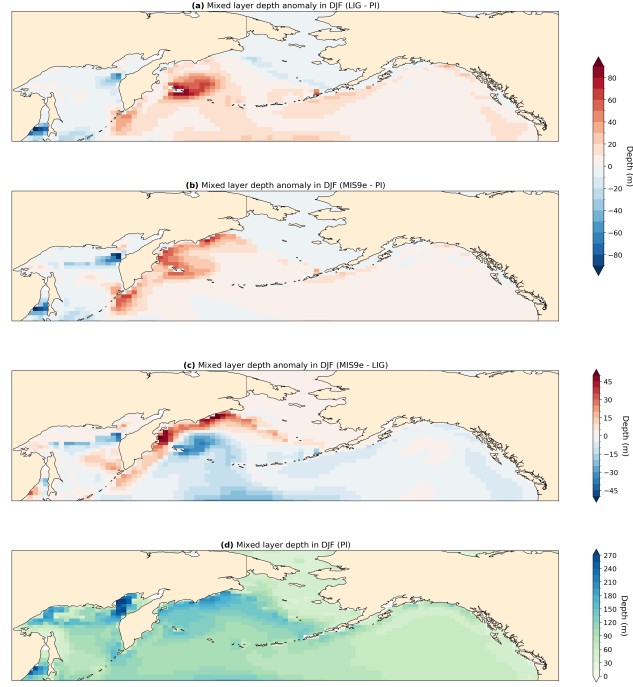

**Figure A8.** Mixed layer depth and anomalies in the North Pacific ocean averaged during boreal winter (DJF) in m. (a) LIG - PI, (b) MIS 9e - PI, (c) MIS 9e - LIG and (d) PI.







**Figure A9.** Water age since last surface contact in years (g) and anomalies (a-f) for LIG - PI (left) and MIS 9e - PI (right). (a, b) 250m depth, (c, d) 500m depth, (e, f) 1000m depth, and (g) 500m PI reference.





**Figure A10.** Water age since last surface contact at 50m depth overlaid with surface wind at 10m (e, f) and anomalies (a-d) off the west coast of North America (left) and off West Africa (right). (a, b) LIG - PI, (c, d) MIS 9e - PI, and (e, f) PI.



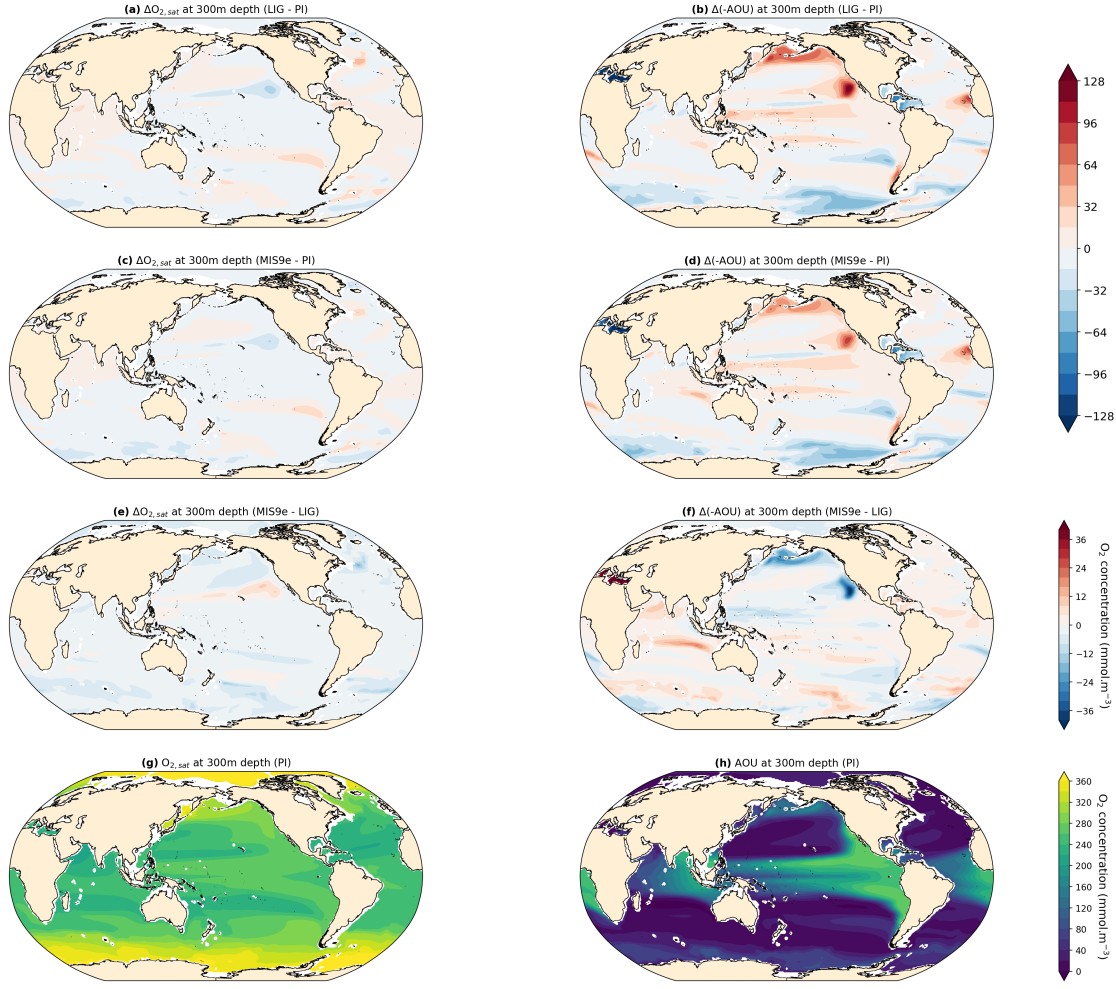

**Figure A11.** Saturated $O_2$ concentration (left) and $-1 \cdot$AOU (right) at 300m depth. Anomalies of LIG - PI (a, b), MIS 9e - PI (c, d) and MIS 9e - LIG (e, f). Full fields for PI are shown in (g, h). To facilitate the comparison between oxygen solubility and oxygen utilization, we multiplied AOU by $-1$.





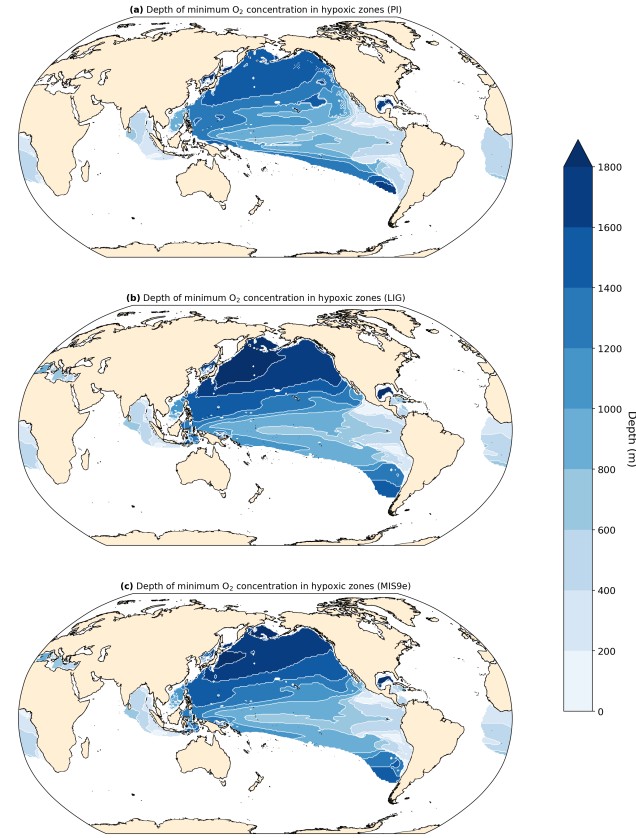

**Figure A12.** Depth of minimum $O_2$ concentration in hypoxic zones ($<62$ mmol·m$^{-3}$). (a) PI, (b) LIG and (c) MIS 9e.



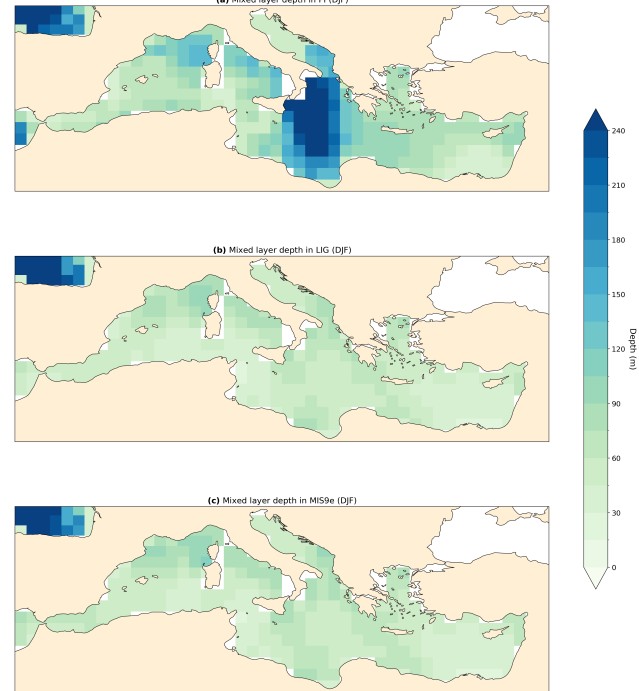

**Figure A13.** Mixed layer depth in the Mediterranean Sea averaged over boreal winter (DJF) in m. (a) PI, (b) LIG and (c) MIS 9e.

*Data availability.* The model data analysed in this manuscript is published on the UNSW ResData repository at
https://doi.org/10.26190/unsworks/30420.

*Author contributions.* BD performed the analyses and drafted the figures under the guidance of KM and LM. KM wrote the manuscript together with BD and LM. NY integrated the LIG and MIS 9e simulations. TZ and MC contributed to the model setup and integrated the PI simulation. BH commented on an advanced draft of the manuscript.

*Competing interests.* LM is a co-editor-in-chief of Climate of the Past. The authors have no other competing interests to declare.

*Disclaimer.* Publisher's note: Copernicus Publications remains neutral with regard to jurisdictional claims in published maps and institutional affiliations



*Acknowledgements.* BD acknowledged funding from the ARC Centre of Excellence for Climate Extremes for an undergraduate intern scholarship. All experiments were performed on the computational facility of National Computational Infrastructure (NCI) owned by the Australian National University through awards under the Merit Allocation Scheme and the UNSW HPC at NCI Scheme. KM and LM acknowledge support from the Australian Research Council (DP180100048, and SR200100008). NY acknowledges the Research Training Program provided by the Australian government, a top-up scholarship provided by the Climate Change Research Centre, and support from the ARC Centre of Excellence for Climate Extremes. TZ and MC receive funding from the Australian Government under the National Environmental Science Program (NESP).



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
