# Peer review of "Simulated ocean oxygenation during the interglacials MIS 5e and MIS 9e"

_EGUsphere, 2024_

## Referee Comment (RC3)

**Overall summary and appreciation**

Duboc et al. present a study on the simulated oxygenation in previous interglacials where their focus is on a comparison of the MIS9e and Last Interglacial (LIG) to the modern (pre-industrial; PI) state. The ACCESS-ESM1.5 model that they employ for their study is a fully coupled atmosphere-ocean model that also considers biogeochemistry in the ocean towards resolving the carbon cycle and related processes that have an effect on oxygenation. The authors present several findings that show that oxygenation of sea water may have looked quite different from today during past interglacials. For me personally, the sensitivity of oxygen availability in the Mediterranean Sea, and in particular the differences that the authors find in this region for the two different interglacials, is the most impressive result. If the authors shared my point of view then I would invite them to reflect in their discussion on some aspects that I suggest below.

The study is certainly of relevance for a wider audience. Understanding the mechanisms that may have impacted on the distribution and extent of oxygen minimum zones during past warm climates may give us an idea of the respective changes that we are currently already facing or that we may have to expect in the future. The question of whether greenhouse gas levels or insolation differences are the most relevant drivers of oxygenation is key in this effort. The authors seem to find that greenhouse gases are less relevant than insolation. I am asking myself whether this statement could be made in a more quantitative manner based on the simulation data at hand, or whether doing so would involve additional simulations that consider a separation of forcings.

I have read the manuscript with great interest and find it generally worth to publish as a research article in Climate of the Past. The text is of high quality, but I also note that graphical presentation of results can still be significantly improved. I would like to ask the authors to also reconsider the overall structure of the article that currently presents various important results in a very extensive appendix.

Please note that while I am a climate modeller, I do not have lots of expertise in sea water oxygenation and in marine biogeochemical cycles. I apologize therefore to the editor and to the authors if I cannot provide very deep insights on these topics.

Please find below various comments that I suggest for consideration towards a revised manuscript.

**General comments:**

I find the results on different oxygenation of the Mediterranean Sea depending on the considered time slice remarkable. In the Mediterranean Sea, the three different interglacials considered (PI, LIG, MIS9e) appear to show completely different response to climate state and boundary conditions. Enhanced stratification (mostly due to reduced surface salinity that is a result of enhanced river runoff to the Mediterranean Sea) certainly contributes to this effect. Could the authors highlight in their discussion differences in the sapropel records between LIG and MIS9e? Sapropels are already summarized in the manuscript, but rather with a focus on different regions than on different time periods. Simulation results shown in Figure 7d indeed show that the Mediterranean Sea is in LIG and MIS9e for most of the year not evaporation controlled, which is the current mode of control, but rather runoff- or preciptation-controlled. Nevertheless, the difference in oxygenation between MIS9e and LIG (Fig. 5c) seems large in comparison to the vertical profiles for temperature, salinity, and water mass age that are shown. Is there a strong non-linearity at play inside the physical / circulation system that could explain this effect, or are there other effects that contribute (e.g. different respiration / activity by simulated organisms in the various interglacials)? Can you make respective statements based on the results derived from the biogeochemistry component of your climate model? Around line 300 of your manuscript you state that imperfect equilibration could be a cause for differences that you find between LIG and MIS9e. Can you give quantitative measures that support this statement? Integration lengths (Table 1) seem rather similar for both simulations. Do you assume that the difference in initial states, or differences in the equilibration times that arise from differences in the forcings, could be the explanation? On the other hand, Figure A11, right column, seems to suggest (although difficult to read) that, while there is more Apparent Oxygen Utilization (AOU; -AOU shown in the graphs, right?) in

LIG and MIS9e than in PI, Last interglacial may see much more utilization of oxygen than MIS9e (I hope that I have correctly applied the factor -1 in this argument). AOU would be due to bioactivity and decomposition of organic material, right? Therefore, some process(es) from the biogeochemistry subsystem of the Earth seem to contribute to the difference in oxygenation between LIG and MIS9e, am I right? Could you discuss in more detail what these contributions could be?

Analysis at orbital time-scales: Are seasonal averages shown in Figures A7, A8, and A13 based on a modern calendar or are seasons corrected for different orbital configuration? If your Fig. A1 is taken as a reference then I assume the calendar is modern (refer to Figure 3 by Otto-Bliesner et al., 2017). If so, please comment on potential implications for your results.

Volume of the appendix in comparison to volume of the manuscript: My observation is that the extent of figures in the appendix (13 figures) is quite large and far beyond the number of figures in the actual manuscript body (7 figures). Various figures in the appendix seem important to the message of the manuscript (in particular Figures A3-A6), and in my humble opinion some of the figures could be considered to present main results of the work by Duboc et al. (in particular A11 and A12). The extensive appendix makes reading the manuscript cumbersome at times, with the need to repeatedly scroll back and forth. I suggest to check whether there is a better way to integrate at least the most important results into the main manuscript text, thereby reducing the size of the appendix.

Presentation of results in figures: I think much more work can be invested in improving figures with regard to clarity and readability. In most cases fonts are far too small to be even barely readable on a normal A4 printout (at least for a person with my eyesight). Proportions of graphical elements are sometimes out of scale (e.g. the vertical spatial extent of the colorbar at Fig. 4 covers about a third to a half of any of the individual figures, which is quite a lot, while, nevertheless, neither the colorbar labels nor the tick labels are readable due to too small font size). The text width of the manuscript, that is available to present graphical results, could be much better utilized by optizing the spacing between individual figure elements (e.g. by reducing the space between left and right columns of Figure 4 and by extending the width of the whole figure panel to cover all useable space of the text width).

Lots of text within figures: As stated above, it is absolutely necessary to increase the font size of text elements in figures across the manuscript. This will of course lead to problems with extensive text information that is included for example as headers of subfigures. I suggest to critically evaluate which part of the information that is currently presented as subfigure heading or annotation must be presented as part of the figure panel, and which part of the information can be savely moved to a figure caption to reduce complexity in the figures themselves.

Spaces between physical units and preceding numbers: In particular for the unit meters there seems to be a more or less consistent lack of spacing between the number and the symbol „m", both in the main text, in Tables, in figure captions, and potentially also in annotations in figures themselves (that I cannot always read). Please check spacing and correct where necessary (e.g. in Table 1, in lines 116, 150, 161, 164, 166, captions of Fig. 1, 3, 4; this list is not exhaustive).

Results on water masses properties, lines 120ff: I was wondering whether it would be sensible to add the main result regarding oxygenation to the title, or to alternatively present it as a clear statement at the end of the respective subsection. Doing so would guide the reader through the most relevant outcomes of your simulations. For example, section title 3.1.2. could become „North Atlantic Deep water - colder and more ventilated".

The metric AOU: Please note that I am not an expert in ocean oxygenation, so I am not very familiar with this metric. Would it be sensible for readers of my low level of expertise to spend a few sentences on the mechanisms that contribute to AOU in reality and in your model? Could you also address in the discussion of your manuscript the differences between AOU and a newer metric True Oxygen Utilisation (Duteil et al., 2013) and outline, to which extend the use of one or the other metric in your study would impact inferences with regard to modelling output or any model-data comparison?

**Specific comments:**

Line 20: should „ocean" be replaced by „oxygen"?

Line 23: model-dependent

Line 79-83: Here I got a bit lost in the formulation. Phytoplankton and zooplanktion represent functional components of the biological system while the aspects described thereafter rather refer to prognostic tracers, is my understanding correct? Would reordering the text as follows improve readability while still conveying correct information? „It includes one functional type of phytoplankton and zooplankton. As prognostic tracers it simulates dissolved inorganic carbon (DIC), alkalinity (ALK), phosphate (PO4), oxygen (O2), and iron." Thereafter, reordering may again improve clarity of formulation: „Detrital decomposition is a function of temperature and is allowed to occur when oxygen is zero. Even though nitrification and denitrification are not explicitly included in the model and the global nitrogen budget is kept constant (Oke et al., 2013), this formulation emulates the effect of denitrification."

Line 84: „All organic and inorganic particles reaching the bottom are remineralized, given that ACCESS-ESM1.5 does not include burial of sediments". Am I right in my assumption that the fact that carbon does not exit the ocean system at the lower boundary, but is instead fed back to the ocean via remineralization, may have an impact on simulated oxygenation of sea water? Please kindly refute or confirm my assumption. If you can confirm my assumption, then please explain (e.g. in your discussion) whether the effect of remineralization on oxygenation is relevant for the inferences that you derive from your study. In particular, is the effect comparable across different climates, so that the observed differences in oxygenation are fully attributable to a difference in climate states rather than a side-effect of the missing sediment module (that may be more (or less) important in dependence of the background climate state).

Line 86: Since Eyring et al. (2016) describe the CMIP6 settings, maybe reformulate: „The pre-industrial 1850 equilibrium simulation (PI) is integrated following the CMIP6 protocol (Eyring et al., 2016) with the exception of using the CMIP5 solar constant (1365.65 W.m−2)."

Line 107: Am I right that a „N/S" should be added after „40°"?

Line 109: Also here please provide the reference(s) (unless Table 1 now clearly lists them).

Line 123-124: „This weakening of deep-ocean convection is mainly due to reduced sea-ice formation" - differences in sea ice formation will impact deep-ocean convection both with regard to the amount of brine rejection and the intensitiy of atmosphere-ocean coupling. Furthermore, presence or absence of sea ice will likely also modulate the atmosphere-ocean gas exchange. Can it be quantified or estimated which of these effects is more relevant?

Line 126: Please define Apparent Oxygen Utilisation. I am not sure whether all readers are familiar with this metric.

Line 130: „with the increase in export production and higher remineralisation rates, and secondarily caused by the temperature-depended solubility"

Line 133: „Figure 2a and d show that"

Line 144: Split the long sentence: „subtropical saline waters. Both are"

Line 145-146: use the abbreviation NADW

Line 168: „at the surface (not shown), that is linked to lower primary productivity and export production"

Line 180: Has significance been tested? If so consider to show this in the figures, e.g. via hatching?

Line 190: For clarity, consider to move the statement „in the LIG simulation" to the start of the sentence.

Line 215: Add commas for clarity of the formulation: „in the MIS9e simulation, dropping to 52.4 mmol m⁻³,"

Line 223: add a space: „128 ka BP", same at line 224 (and potentially at other locations)

Line 289: try to avoid the word segmenta of MIS 9e being separated by line breaks (maybe do not use a space in the term, across your manuscript?)

Line 302: This region remains

Line 308: please check whether „dependent on" or „dependent of" is grammatically correct here (I am not native English speaker)

Line 312: long equilibration times?

Line 317-319: Is your statement regarding higher sensitivity to insolation than to greenhouse gases robust if also taking into account the differences in oxygenation found between interglacials in the Mediterranean? I think that a strict discrimination between different sensitivties and estimation of their relative strength would be only possible based on simulations that include separations of forcing, am I right?

Line 330: „and 63% is hypoxic"

Line 335: consider the spelling of skillful, at least at one other location you spell it with one „l"

**Specific comments to Figures (beyond font size, which represents a problem across the manuscript) and Tables:**

Table 1: Please provide the reference to the parameter values provided, in particular orbital parameter solution and reference for greenhouse gases. While on may guess that Otto-Bliesner et al. (2017) is the reference for PI and LIG, providing the citation is particularly important for MIS 9e where there is no obvious reference that comes to my mind. Such information may also be used to make the table heading a bit more informative. Regarding footnote 1: How, if at all, does erroneous forcing affect results? It is appreciated that such information is conveyed to the reader. Nevertheless, a bit of evaluation on the potential impacts (or the absence of such) may be of interest to the readers.

Figure 2: Please define AOU in the caption (also in other Figure captions, where needed). Contour lines and contour line labels carry little information for me since I cannot read them.

Figure 4: Differences in black contour lines (solid, dotted, dashed) not visible to me. Furthermore, I am not sure about usage of footnotes in figure captions. I would avoid them and rather implement the information directly as caption text.

Figure A1: Show hovmöller plots with calendar corrected data, unless there is a good reason to use the modern calendar.

Figure A2: It is difficult for me to extract any meaningful information from the very small legends, boxes, and thin lines. Keeping this in mind, it is also difficult for me to identify any interpolated data that you refer to in the caption.

Figure A5: Can you explain the artifact at about 20°N in the AMOC (left column)? My first guess would have been the Strait of Gibraltar, but fluxes across the gateway are probably too small and the location of the artifact also does not fit.

Figure A6: There is stronger AMOC in LIG and MIS9e (Fig. A5) but the mixed layer depths are smaller in these simulations than they are in PI; i.e., there is a weaker link between AMOC strength and mixed layer depth in MIS9e and LIG. Does this difference in dynamics contribute to your findings regarding oxygenation? Furthermore, you mention in the main text that changes in deep mixing are linked to different sea ice conditions. So, can the weakened deep convection in the Barents Sea near Svalbard be explained by less autumn-to-winter sea ice formation and the related reduced brine rejection?

Figure A8: Would it make sense to also show the PMOC to understand the links between deep ventilation and meridional overturning?

Figure A9: Is there a reason why you compare in a and b different depths with each other (250 m vs 50 m)?

Figure A11: define the abbreviation AOU

**References:**

Eyring, V., Bony, S., Meehl, G. A., Senior, C. A., Stevens, B., Stouffer, R. J., and Taylor, K. E.: Overview of the Coupled Model Intercomparison Project Phase 6 (CMIP6) experimental design and organization, Geosci. Model Dev., 9, 1937–1958, https://doi.org/10.5194/gmd-9-1937-2016, 2016.

Otto-Bliesner, B. L., Braconnot, P., Harrison, S. P., Lunt, D. J., Abe-Ouchi, A., Albani, S., Bartlein, P. J., Capron, E., Carlson, A. E., Dutton, A., Fischer, H., Goelzer, H., Govin, A., Haywood, A., Joos, F., LeGrande, A. N., Lipscomb, W. H., Lohmann, G., Mahowald, N., Nehrbass-Ahles, C., Pausata, F. S. R., Peterschmitt, J.-Y., Phipps, S. J., Renssen, H., and Zhang, Q.: The PMIP4 contribution to CMIP6 – Part 2: Two interglacials, scientific objective and experimental design for Holocene and Last Interglacial simulations, Geosci. Model Dev., 10, 3979–4003, https://doi.org/10.5194/gmd-10-3979-2017, 2017.

Duteil, O., Koeve, W., Oschlies, A., Bianchi, D., Galbraith, E., Kriest, I., and Matear, R.: A novel estimate of ocean oxygen utilisation points to a reduced rate of respiration in the ocean interior, Biogeosciences, 10, 7723–7738, https://doi.org/10.5194/bg-10-7723-2013, 2013.

---

## Author Comment (AC1)

Reviewer 1

We would like to thank Reviewer 1 for their thoughtful and helpful review. We have replicated the Reviewer's comments below in blue and italics. Our responses to each of the comments are in black and changes to the text in the manuscript are in red and bold. Thank you for your time and expertise!

*The study focuses on oxygenation during past interglacial periods, showing clear results on oxygenation patterns and differences between MIS 5e and MIS 9e in terms of ocean oxygenation. I recommend that the paper be accepted for publication with minor revisions.*

*1 When comparing the PI and global distributions of dissolved O2 concentration, did you account for the influence of greenhouse gases increase? Maybe this comparison can further elucidate the effects of greenhouse gases. Is there any simulation conducted with modern atmospheric concentrations?*

Thank you for this suggestion. We have indeed a transient historical simulation from 1850 to 2015. The WOA data is a compilation of observations spanning 1965-2022. Below we show the vertical mean $O_2$ concentration for our PI simulation in (a); for our historical run, averaged between 1965 and 2015, in (b); and the WOA data (1965-2022) in (c).

The difference between the PI simulation and the historical run is very small; the change in greenhouse gases in this transient simulation therefore does not change the fit with observations significantly.

[Figure]

*Fig R1.1 Vertically averaged dissolved $O_2$ concentrations (mmol/m$^3$). (a) PI simulation ; (b) historical simulation, averaged over 1965-2015; (c) WOA data (1965-2022).*

We therefore decided not to change Figure 1 (new Figure 3) in the paper.

We have amended the sentence discussing the impact of changes in greenhouse gas concentrations in the first paragraph of the discussion as follows:

"**The large-scale ocean circulation patterns, including AABW, NADW and the ventilation of the North Pacific Ocean are therefore very sensitive to the latitudinal and seasonal distribution of incoming solar radiation in the ACCESS ESM1.5, and less sensitive to changes in greenhouse gas concentrations within the range of these three interglacials.**"

*2 The article does not perform a significance test when calculating anomalies, which should be included.*

We would like to thank the Reviewer for this suggestion (also mentioned by Reviewer 2). We have now performed significance tests on all results. All figures that show anomalies now clearly indicate regions that are not statistically significant.

*3 Some of the text in the figures is too small to read clearly (e.g. titles in Fig. 2,3,4..). Please increase the font size to improve readability.*

This comment was mirrored by Reviewers 2 and 3. We have now changed all the figures. In particular we have:

- Removed the titles of the subplots;
- Made the legend colour bars thinner where appropriate;
- Increased the font size for all remaining text in the figures;
- Reduced the white space between subpanels where possible.

*4 Certain figures appear unnecessary. I suggest the authors to reduce the number of figures in appendix.*

Reviewer 3 suggested moving a few key figures from the Appendix to the main text. We have now moved A2, A3 and A5 to the main text. This has reduced the number of figures in the Appendix to ten.

*5 More discussion about how ocean oxygen may change in the context of increasing carbon dioxide concentrations in the future and the potential impacts of such changes. I hope these suggestions assist the authors in revising the manuscript.*

Following paragraph has been added to Section 4:

"**Our study shows that even relatively small changes in boundary conditions can lead to large changes in ocean circulation, upwelling systems, export production, and ocean oxygenation. While our results cannot directly inform on future changes in ocean oxygenation, there might be some similarities. Ocean temperatures and ocean stratification are projected to continue to increase until atmospheric $CO_2$ concentrations finally plateau. Current ocean deoxygenation is therefore not easily reversible and will persist for centuries (Oschlies, 2021). There will be physiological and morphological impacts on organisms, including reduced growth for a vast range of taxonomic groups (Sampaio et al., 2021). Exposure to low oxygen conditions has also been associated with a delay when fish produce eggs, a reduction of the number of eggs fish produce and blindness (Landry et al., 2007; McCormick andLevin,**

**2077). The metabolic demand of oxygen increases with water temperature, and when combined with deoxygenation, this can lead to respiratory distress, followed by respiratory failure and death (Clarke et al., 2021). Over 50 mass mortality events due to hypoxia have been recorded in the tropics to date (Altieri et al., 2017). The consequences of deoxygenation for fisheries and the world's future food supply could thus be serious (Oschlies et al., 2018; Rose et al., 2019)."**

Thank you again for the very helpful review that helped us improve the paper.

---

## Author Comment (AC2)

We would like to thank Reviewer 2 for their insightful and helpful review. We have replicated the Reviewer's comments below in blue and italics. Our responses to each of the comments are in black and changes to the text in the manuscript are in red and bold. Thank you for your time and expertise!

*The study by Duboc et al. investigated ocean oxygenation during the MIS 5e and MIS9e based on ACCESS-ESM1.5. The topic fits the CP, but the organization of the manuscript needs to improve.*

*Major comments:*

1. *In the Introduction section, the authors provide reconstructed ocean oxygenation in the past climate using considerable length, and I think this paragraph can be summarized more briefly. Moreover, as this manuscript focused on the modeling, I recommend providing some key points on research progress on the numerical modeling. In addition, I suggest adding a paragraph at the end of this section by giving the organization of this manuscript.*

To the authors' knowledge, this is the first study modelling ocean oxygen conditions for recent interglacials. It is therefore not possible to provide more background on other modelling studies and discuss research progress on numerical modelling of oxygen during interglacials. We therefore decided not to shorten the paragraph in the introduction that discusses oxygenation during interglacials based on proxy reconstructions.

We have added following paragraph to the end of Section 1:

**"This manuscript is structured as follows. Section 2 describes the model, the experimental setup and the time series for each of the analyzed experiments. Section 3 first analyses changes in large-scale circulation patterns and oxygenation and then focuses on changes in the Mediterranean Sea. In Section 4 we discuss uncertainties and situate our results within a broader context. Section 5 summarizes the main results."**

2. *Lines 47-53: I feel confused about this paragraph. Why do you separate the Mediterranean Sea individually? If necessary, please provide transition sentences.*

We agree that a better transition was needed here and have changed the sentence to:

**"One region with notably large changes in past ocean oxygen concentrations is the semi-enclosed Mediterranean Sea, where there is evidence of intervals with severe anoxia during past interglacials (Sachs and Repeta, 1999; Rohling et al., 2015; Rush et al., 2019). These intervals are characterised by…"**

3. *In Section 2, the authors provide 1000~2000 model-year simulations in this study. In Figure A2, I don't think the simulated conditions reaching quasi-equilibrium. It is obvious that the global mean $O_2$ concentration, mean $O_2$ concentration in the Mediterranean Sea, and AABW mean $O_2$ still show a decreasing trend. Indeed, the deep ocean circulation generally needs more than 4000 model-year integration to reach quasi-equilibrium. I am wondering whether this imbalance may have an impact on your results.*

There is indeed still a small drift in O₂ in the global mean and in the Mediterranean Sea. Please see below zonal mean O₂ concentration drifts (shown here as anomalies between the last two 250-year averages).

[Figure]

*Figure R2.1: Zonal-mean oxygen concentration anomalies (250-year averages). 250 last years of simulation minus 250 previous years.*

We have now moved Figure A2 to the main text (new Figure 1) and discussed this throughout the manuscript:

Methods:

**"Dissolved oxygen is a tracer with very long equilibration times. Figure 1 shows time series of dissolved oxygen for the three simulations analysed in this study. While oxygen concentrations have reached quasi-equilibrium for North Atlantic Deep Water (NADW), in the intermediate waters of the North Pacific, and in Antarctic Bottom Water south of 35ºS in all runs, there is still a slight drift in global mean dissolved oxygen in our LIG and MIS 9e simulations and a more substantial drift in the Mediterranean Sea."**

Results:

**"It should be noted that oxygen concentrations in the deep waters in the Ionian Sea are still drifting in our simulations and might reach suboxic values if integrated for longer."**

**"It should be noted that oxygen levels in the Mediterranean Sea are still drifting in both simulations and that the equilibrium values are lower than what is presented here."**

Discussion:

**"In addition, note that Mediterranean oxygen content is still drifting in our LIG and MIS 9e runs. As a result, MIS 9e could reach similar oxygen loss as LIG on a longer timescale (Figure 1)."**

Conclusions:

**"Dissolved oxygen concentrations are still drifting in both simulations, even after integration times of well over 1,500 years."**

"Oceanic oxygen takes centuries to adjust, and we currently lack computer power to run high-resolution models for the timescales needed for initialisation and equilibrium responses."

4. *In Section 3.1, the authors provide oxygenation changes in AABW, NADW, NPIW, and OMZ. Why do you choose these water/regions for analysis? Any opinions on other water masses? At least, the authors should introduce the importance of these water masses at the beginning of this section.*

We have added the following paragraph to the end of Section 3.1.:

"The next two subsections analyze changes in the main large-scale bottom and deep water masses, AABW and NADW. We also see significant changes in the oxygenation of North Pacific Intermediate Water, which are described in Section 3.1.3. Changes in low oxygen zones can have a large impact on marine life and biogeochemical cycles. We therefore analyze changes in the simulated Oxygen Minimum Zones in Section 3.1.4."

5. *In Section 3.1.1, authors declare changes in AOU, export production, and remineralization rates during the MIS 5e and MIS 9e. However, readers may feel confused about how these factors influence ocean oxygenation, because not all readers are familiar with these words. I suggest adding some illustrations on the definition and impact of AOU, export production and remineralization rates.*

We apologize for the omission. Following paragraph has been added to the end of Section 2:

"In Section 3 we partition changes in dissolved oxygen into two components, changes in the saturated concentration of oxygen $O_2^{sat}$ and changes in Apparent Oxygen Utilisation (AOU). AOU estimates the oxygen consumed during respiration and can be calculated as the difference of dissolved oxygen concentration ($O_2$) and $O_2^{sat}$:

$$AOU = O_2^{sat}(T, S) - O_2 \qquad\qquad (1)$$

where T is the potential temperature and S salinity. Changes in AOU are therefore a combination of changes in circulation (with sluggish water masses tending to have higher AOU), and changes in remineralisation rates, which depend on the vertical export of organic matter (export production) and temperature. Please note that the here used metric AOU assumes that dissolved oxygen in surface waters is in equilibrium with the atmosphere, which might lead to an overestimation of the True Oxygen Utilisation (TOU) (Duteil et al., 2013)."

6. *Following comment #4, in Section 3.1.1, the authors declare changes in dissolved oxygen concentrations in AABW are linked with changes in ocean circulations, with the increase in export production and remineralization rates. Do changes in ocean circulation impact export production and remineralization rates? If so, how do you distinguish their exact contribution to the ocean oxygen concentrations? Moreover, ocean circulation, export production, and remineralization rates are linked with increased temperature. Therefore, I speculate that changes in thermal structures in the ocean induced by solar insolation anomalies may drive variations in ocean oxygenation ultimately. Same comments on Section 3.1.2-3.1.4.*

Unfortunately, we cannot distinguish between the contribution of circulation changes, export production and remineralization rates to changes in oxygen concentrations within this modelling frame.

We can calculate saturated $O_2$, which depends on water temperature and salinity. The difference between in situ $O_2$ and saturated $O_2$ then gives us an indication of $O_2$ consumption.

$O_2$ consumption, or apparent oxygen utilization, for a particular water parcel will depend on the history of this water parcel since it has last "seen" the atmosphere. It will be the integral of remineralisation rates during its journey. When we compare two simulations, changes in remineralisation rates can be therefore be due to

(a) changes in the availability of organic matter to remineralise on the pathway of this water parcel (i.e. changes in export production),

(b) changes in temperature on the pathway of this water parcel (i.e. remineralisation will increase with higher temperatures due to higher metabolic rates), and, most importantly,

(c) changes in circulation (i.e., if circulation weakens and residence times increase, the integral of remineralisation rates since last contact with the atmosphere will increase).

In addition, ocean circulation changes will of course also be influenced by temperature changes, and export production will also depend on nutrient availability (and temperature), which depends on ocean circulation.

7. *Following comment #1, in Section 3.2, I am curious about the uniqueness of the Mediterranean Sea. I suggest adding some sentences in the Introduction and Section 3 that illustrate the necessity of separating the results of the Mediterranean Sea individually, e.g. its large region, substantial drift in oxygenation…*

We have rephrased the sentence introducing the Mediterranean Sea in the Introduction as follows:

**"One region with notably large changes in past ocean oxygen concentrations is the semi-enclosed Mediterranean Sea, where there is evidence of intervals with severe anoxia during past interglacials (Sachs and Repeta, 1999; Rohling et al., 2015; Rush et al., 2019). These intervals are characterised by…"**

We added following paragraph at the beginning of Section 3.2.:

**"The Mediterranean Sea is a semi-enclosed sea which is linked to the Atlantic Ocean via the relatively narrow and shallow (< 900 m) Strait of Gibraltar. Changes in large-scale circulation therefore impact the Mediterranean Sea only moderately, although the opposite is not true, as the Mediterranean outflow impacts the large-scale circulation in the Atlantic Ocean (Barbosa Aguiar et al. , 2015). There have been numerous time intervals during past interglacials when some regions of the Mediterranean Sea became anoxic, prompting us to analyse changes in simulated dissolved $O_2$ in the Mediterranean Sea in this section."**

8. *Lines 246-250. How is this conclusion proposed? Indeed, although Earth's orbit and greenhouse gases are the main external forcings in MIS 5e and MIS 9e. However, changes in incoming solar insolation are the dominant external forcings, with its radiative forcing quite larger than the*

*greenhouse gases. I speculate that the less oxygenated conditions during MIS 5e and MIS 9e may be tied to changes in solar insolation driven by Earth's orbit compared with the PI, rather than their differences in greenhouse gases.*

Yes, the Reviewer is correct. To make the conclusion clearer, we added the words in red:

"**The large-scale ocean circulation patterns, including AABW, NADW and the ventilation of the North Pacific Ocean are therefore very sensitive to the latitudinal and seasonal distribution of incoming solar radiation in the ACCESS ESM1.5, and less sensitive to changes in greenhouse gas concentrations within the range of these three interglacials. The insolation anomalies are indeed quite similar for LIG and MIS 9e (Figure A1), while the greenhouse gas forcing is highest for MIS 9e, lowest for LIG, with PI being in the middle (Table 1).**"

9. *In the Abstract and Introduction sections, the authors illustrated that the MIS 5e and MIS 9e may provide a reference for ocean oxygenation in a warmer world. I want to see more discussions on how the less oxygenated conditions during MIS 5e and MIS 9e may provide insight for reducing uncertainties in future ocean deoxygenation.*

Following paragraph has been added to Section 4:

"**Our study shows that even relatively small changes in boundary conditions can lead to large changes in ocean circulation, upwelling systems, export production, and ocean oxygenation. While our results cannot directly inform on future changes in ocean oxygenation, there might be some similarities. Ocean temperatures and ocean stratification are projected to continue to increase until atmospheric $CO_2$ concentrations finally plateau. Current ocean deoxygenation is therefore not easily reversible and will persist for centuries (Oschlies, 2021). There will be physiological and morphological impacts on organisms, including reduced growth for a vast range of taxonomic groups (Sampaio et al., 2021). Exposure to low oxygen conditions has also been associated with a delay when fish produce eggs, a reduction of the number of eggs fish produce and blindness (Landry et al., 2007; McCormick andLevin, 2077). The metabolic demand of oxygen increases with water temperature, and when combined with deoxygenation, this can lead to respiratory distress, followed by respiratory failure and death (Clarke et al., 2021). Over 50 mass mortality events due to hypoxia have been recorded in the tropics to date (Altieri et al., 2017). The consequences of deoxygenation for fisheries and the world's future food supply could thus be serious (Oschlies et al., 2018; Rose et al., 2019).**"

*Minor comments:*

1. *Line 160, adding abbreviation (OMZ).*

The abbreviation was added.

2. *Fig. 2, adding boxes indicating the region of AABW, NADW….*

We decided not to add these boxes, as this visualisation is standard in oceanography and boxes would reduce the visibility of the plots.

3. *Adding significant test for anomalies between MIS 5e and PI, MIS 9e and PI, and MIS 9e and MIS 5e.*

We would like to thank the Reviewer for this suggestion (also mentioned by Reviewer 1). We have now performed significance tests on all results. All figures that show anomalies now clearly indicate regions that are not statistically significant.

4. *Enlarging the titles in the figures, they are too small.*

This comment was mirrored by Reviewers 1 and 3. We have now changed all the figures. In particular, we have:

- Removed the titles of the subplots;
- Made the legend colour bars thinner where appropriate;
- Increased the font size for all remaining text in the figures;
- Reduced the white space between subpanels where possible.

Thank you again for the helpful review that helped us improve the paper.

---

## Author Comment (AC3)

*Overall summary and appreciation*

*Duboc et al. present a study on the simulated oxygenation in previous interglacials where their focus is on a comparison of the MIS9e and Last Interglacial (LIG) to the modern (pre-industrial; PI) state. The ACCESS-ESM1.5 model that they employ for their study is a fully coupled atmosphere-ocean model that also considers biogeochemistry in the ocean towards resolving the carbon cycle and related processes that have an effect on oxygenation. The authors present several findings that show that oxygenation of sea water may have looked quite different from today during past interglacials. For me personally, the sensitivity of oxygen availability in the Mediterranean Sea, and in particular the differences that the authors find in this region for the two different interglacials, is the most impressive result. If the authors shared my point of view then I would invite them to reflect in their discussion on some aspects that I suggest below.*

We would like to thank the Reviewer for this judgment, and we agree.

*The study is certainly of relevance for a wider audience. Understanding the mechanisms that may have impacted on the distribution and extent of oxygen minimum zones during past warm climates may give us an idea of the respective changes that we are currently already facing or that we may have to expect in the future. The question of whether greenhouse gas levels or insolation differences are the most relevant drivers of oxygenation is key in this effort. The authors seem to find that greenhouse gases are less relevant than insolation. I am asking myself whether this statement could be made in a more quantitative manner based on the simulation data at hand, or whether doing so would involve additional simulations that consider a separation of forcings.*

Unfortunately, a more quantitative approach would require additional sensitivity simulations which would be very expensive and long to run. We would also like to point out that this statement is only valid for the relatively small changes in greenhouse gas forcing between these three interglacials. We have now rephrased this statement to:

"**The large-scale ocean circulation patterns, including AABW, NADW and the ventilation of the North Pacific Ocean are therefore very sensitive to the latitudinal and seasonal distribution of incoming solar radiation in the ACCESS ESM1.5, and less sensitive to changes in greenhouse gas concentrations within the range of these three interglacials.**"

*I have read the manuscript with great interest and find it generally worth to publish as a research article in Climate of the Past. The text is of high quality, but I also note that graphical presentation of results can still be significantly improved. I would like to ask the authors to also reconsider the overall structure of the article that currently presents various important results in a very extensive appendix.*

We agree and we have made the requested changes (see our responses below).

*Please note that while I am a climate modeller, I do not have lots of expertise in sea water oxygenation and in marine biogeochemical cycles. I apologize therefore to the editor and to the authors if I cannot provide very deep insights on these topics.*

*Please find below various comments that I suggest for consideration towards a revised manuscript.*

*General comments:*

*I find the results on different oxygenation of the Mediterranean Sea depending on the considered time slice remarkable. In the Mediterranean Sea, the three different interglacials considered (PI, LIG, MIS9e) appear to show completely different response to climate state and boundary conditions. Enhanced stratification (mostly due to reduced surface salinity that is a result of enhanced river runoff to the Mediterranean Sea) certainly contributes to this effect. Could the authors highlight in their discussion differences in the sapropel records between LIG and MIS9e?*

That is an excellent idea, but unfortunately there isn't that much work that compares the different sapropels in the Mediterranean either qualitatively or quantitatively in terms of oxygenation, hydrology, primary productivity. There are suggestions that the transition from dry to wet conditions during sapropel events was different between S5 (LIG) and S10 (MIS 9e), with the latter changes being rapid compared to progressive under S5 (Melki et al., 2010), but whether this influenced seawater oxygen contents is unclear.

*Sapropels are already summarized in the manuscript, but rather with a focus on different regions than on different time periods. Simulation results shown in Figure 7d indeed show that the Mediterranean Sea is in LIG and MIS9e for most of the year not evaporation controlled, which is the current mode of control, but rather runoff- or precipitation-controlled. Nevertheless, the difference in oxygenation between MIS9e and LIG (Fig. 5c) seems large in comparison to the vertical profiles for temperature, salinity, and water mass age that are shown. Is there a strong non-linearity at play inside the physical / circulation system that could explain this effect, or are there other effects that contribute (e.g. different respiration / activity by simulated organisms in the various interglacials)? Can you make respective statements based on the results derived from the biogeochemistry component of your climate model?*

The rate of change in $O_2$ is similar for LIG and MIS 9e (new Figure 1b), although a bit steeper for the LIG simulation. The main difference in oxygenation between these two simulations is therefore due to the fact that the LIG stratifies earlier and has spent more time in a stratified state (please also see our response to the next comment).

However, there are indeed differences in biological productivity between LIG and MIS 9e which could explain the difference in the slopes in Figure 1b. Fig R3.1. shows that phytoplankton concentrations in the MIS 9e simulation are smaller than in the LIG simulation, especially in the Sea of Sicily. This is due to higher nutrient availability (Fig R3.2.) in the LIG simulation which is caused by slightly enhanced mixing off the coast of North Africa.

[Figure]

[Figure]

[Figure]

*Figure R3.1. Phytoplankton concentration averaged over the top 50 m in the Mediterranean Sea for (a) LIG, (b) MIS 9e, (c) MIS9e-LIG.*

[Figure]

*Figure R3.2. Nitrate concentration averaged over the top 50 m in the Mediterranean Sea for (a) LIG, (b) MIS 9e, (c) MIS9e-LIG.*

[Figure]

Ocean temperatures are comparable for these simulations in the Mediterranean Sea (new Figure 8a), so the temperature-dependent remineralisation rates should be comparable.

We have added following sentence to Section 3.2:

"**Sea surface salinities decrease at a faster rate and lead to stratification and deoxygenation earlier than in the MIS 9e simulation (Figure 1b). The difference in oxygenation between LIG and MIS 9e is thus mainly due to the difference in the length of time during which the Mediterranean Sea is stratified, although small changes in biological productivity and export production (not shown) also contribute. It should be noted that oxygen levels in the Mediterranean Sea are still drifting in both simulations and that the equilibrium values are lower than what is presented here.**"

*Around line 300 of your manuscript you state that imperfect equilibration could be a cause for differences that you find between LIG and MIS9e. Can you give quantitative measures that support this statement? Integration lengths (Table 1) seem rather similar for both simulations. Do you assume that the difference in initial states, or differences in the equilibration times that arise from differences in the forcings, could be the explanation?*

New Figure 1b shows the time series of oxygen content in the Mediterranean Sea for the three simulations. It can be seen that the LIG and MIS9e simulations follow the PI control simulation closely for the first ~300 years. At around ~300 years, oxygen starts to decline in the LIG simulation. The increased river runoff therefore needs a few hundred years to accumulate before the Mediterranean Sea stratifies and oxygen at depth starts to decline. The same is true for the MIS9e simulation, but it branches off later, after ~500 years. This is due to the fact that the monsoon is not quite as strong as in the LIG simulation (new Fig 10). It therefore takes longer for the Mediterranean Sea to stratify in the MIS9e simulation than in the LIG simulation. Once stratified, oxygen declines at an almost linear rate in both simulations, although the LIG simulation shows some signs of leveling off, nearing equilibration.

It is unfortunately impossible to give a quantitative statement on where the MIS9e simulation will level off without extending the simulations.

*On the other hand, Figure A11, right column, seems to suggest (although difficult to read) that, while there is more Apparent Oxygen Utilization (AOU; -AOU shown in the graphs, right?) in LIG and MIS9e than in PI, Last interglacial may see much more utilization of oxygen than MIS9e (I hope that I have correctly applied the factor -1 in this argument). AOU would be due to bioactivity and decomposition of organic material, right? Therefore, some process(es) from the biogeochemistry subsystem of the Earth seem to contribute to the difference in oxygenation between LIG and MIS9e, am I right? Could you discuss in more detail what these contributions could be?*

The AOU is calculated as the difference between saturated $O_2$ and in situ $O_2$. It is  therefore a diagnostic of oxygen consumption. For a particular water parcel, this consumption will depend on the history of this water parcel since it has last "seen" the atmosphere. It will be the integral of remineralisation rates during its journey. Changes in AOU can therefore be due to:

(a) changes in the availability of organic matter to remineralise on the pathway of this water parcel (i.e. changes in export production),

(b) changes in temperatures on the pathway of this water parcel (i.e. remineralisation will increase with higher temperatures due to higher metabolic rates), and, most importantly,

(c) changes in circulation (i.e., if circulation weakens and residence times increase, the integral of remineralisation rates since last contact with the atmosphere will increase).

For the Mediterranean Sea, these large increases in AOU are mainly due to changes in circulation. They would be expected to be higher for the LIG run than for the MIS9e run, because the LIG run stratified earlier in the simulation, and the stratification is stronger.

Biological productivity and export production also plays a role, as biological productivity is slightly reduced at MIS 9e compared to LIG (Fig R3.1.c). It is unfortunately impossible to disentangle these processes in the current modelling frame.

As mentioned above, we have added following sentence to Section 3.2:

"**Sea surface salinities decrease at a faster rate and lead to stratification and deoxygenation earlier than in the MIS 9e simulation (Figure 1b). The difference in oxygenation between LIG and MIS 9e is thus mainly due to differences in the length of time during which the Mediterranean Sea is stratified, although small changes in biological productivity and export production (not shown) also contribute. It should be noted that oxygen levels in the Mediterranean Sea are still drifting in both simulations and that the equilibrium values are lower than what is presented here.**"

*Analysis at orbital time-scales: Are seasonal averages shown in Figures A7, A8, and A13 based on a modern calendar or are seasons corrected for different orbital configuration? If your Fig. A1 is taken as a reference then I assume the calendar is modern (refer to Figure 3 by Otto-Bliesner et al., 2017). If so, please comment on potential implications for your results.*

All seasonal plots have now been replotted with the LIG adjusted calendar. However, we have not changed Figure A1 because we prefer a linear scale for the x-axis of this Hovmöller diagram. The x-axis of A1 therefore remains as "day of the year" without artificially expanding or contracting time due to differences in lengths of different months.

*Volume of the appendix in comparison to volume of the manuscript: My observation is that the extent of figures in the appendix (13 figures) is quite large and far beyond the number of figures in the actual manuscript body (7 figures). Various figures in the appendix seem important to the message of the manuscript (in particular Figures A3-A6), and in my humble opinion some of the figures could be considered to present main results of the work by Duboc et al. (in particular A11 and A12). The extensive appendix makes reading the manuscript cumbersome at times, with the need to repeatedly scroll back and forth. I suggest to check whether there is a better way to integrate at least the most important results into the main manuscript text, thereby reducing the size of the appendix.*

We have now moved Figures A2, A3 and A5 to the main text.

*Presentation of results in figures: I think much more work can be invested in improving figures with regard to clarity and readability. In most cases fonts are far too small to be even barely readable on a normal A4 printout (at least for a person with my eyesight). Proportions of graphical elements are sometimes out of*

*scale (e.g. the vertical spatial extent of the colorbar at Fig. 4 covers about a third to a half of any of the individual figures, which is quite a lot, while, nevertheless, neither the colorbar labels nor the tick labels are readable due to too small font size). The text width of the manuscript, that is available to present graphical results, could be much better utilized by optizing the spacing between individual figure elements (e.g. by reducing the space between left and right columns of Figure 4 and by extending the width of the whole figure panel to cover all useable space of the text width).*

*Lots of text within figures: As stated above, it is absolutely necessary to increase the font size of text elements in figures across the manuscript. This will of course lead to problems with extensive text information that is included for example as headers of subfigures. I suggest to critically evaluate which part of the information that is currently presented as subfigure heading or annotation must be presented as part of the figure panel, and which part of the information can be savely moved to a figure caption to reduce complexity in the figures themselves.*

This comment was mirrored by Reviewers 1 and 2. We have now changed all the figures. In particular, we have:

- Removed the titles of the subplots;
- Made the legend colour bars thinner where appropriate;
- Increased the font size for all remaining text in the figures;
- Reduced the white space between subpanels where possible.

*Spaces between physical units and preceding numbers: In particular for the unit meters there seems to be a more or less consistent lack of spacing between the number and the symbol „m", both in the main text, in Tables, in figure captions, and potentially also in annotations in figures themselves (that I cannot always read). Please check spacing and correct where necessary (e.g. in Table 1, in lines 116, 150, 161, 164, 166, captions of Fig. 1, 3, 4; this list is not exhaustive).*

This has been fixed.

*Results on water masses properties, lines 120ff: I was wondering whether it would be sensible to add the main result regarding oxygenation to the title, or to alternatively present it as a clear statement at the end of the respective subsection. Doing so would guide the reader through the most relevant outcomes of your simulations. For example, section title 3.1.2. could become „North Atlantic Deep water - colder and more ventilated".*

We have changed the subtitles as follows:

**"3.1.1 Antarctic Bottom Water - warmer and less ventilated"**

**"3.1.2 North Atlantic Deep Water - colder and better ventilated"**

**"3.1.3 North Pacific Intermediate Water - warmer and better ventilated"**

**"3.1.4 Oxygen Minimum Zones (OMZs) - no significant change"**

**"3.2 Oxygenation of the Mediterranean Sea - large-scale hypoxia"**

*The metric AOU: Please note that I am not an expert in ocean oxygenation, so I am not very familiar with this metric. Would it be sensible for readers of my low level of expertise to spend a few sentences on the mechanisms that contribute to AOU in reality and in your model? Could you also address in the discussion of your manuscript the differences between AOU and a newer metric True Oxygen Utilisation (Duteil et al., 2013) and outline, to which extend the use of one or the other metric in your study would impact inferences with regard to modelling output or any model-data comparison?*

This was also flagged by Reviewer 2 and we apologize for the omission. Following paragraph has been added to the end of Section 2:

**"In Section 3 we partition changes in dissolved oxygen into two components, changes in the saturated concentration of oxygen $O_2^{sat}$ and changes in Apparent Oxygen Utilisation (AOU). AOU estimates the oxygen consumed during respiration and can be calculated as the difference of dissolved oxygen concentration ($O_2$) and $O_2^{sat}$:**

**$AOU = O_2^{sat}(T, S) - O_2$          (1)**

**where T is the potential temperature and S salinity. Changes in AOU are therefore a combination of changes in circulation (with sluggish water masses tending to have higher AOU), and changes in remineralisation rates, which depend on the vertical export of organic matter (export production) and temperature. Please note that the here used metric AOU assumes that dissolved oxygen in surface waters is in equilibrium with the atmosphere, which might lead to an overestimation of the True Oxygen Utilisation (TOU) (Duteil et al., 2013)."**

*Specific comments:*

*Line 20: should „ocean" be replaced by „oxygen"?*

Nice catch! Thank you.

*Line 23: model-dependent*

"dependent" is correct in American spelling, whereas "dependant" is correct in British and Australian spelling. Given that we are Australian, we left the text in Australian (British) spelling for now - and Climate of the Past can then change the spelling as they see fit according to their publication guidelines.

*Line 79-83: Here I got a bit lost in the formulation. Phytoplankton and zooplanktion represent functional components of the biological system while the aspects described thereafter rather refer to prognostic tracers, is my understanding correct? Would reordering the text as follows improve readability while still conveying correct information? „It includes one functional type of phytoplankton and zooplankton. As prognostic tracers it simulates dissolved inorganic carbon (DIC), alkalinity (ALK), phosphate (PO4), oxygen (O2), and iron." Thereafter, reordering may again improve clarity of formulation: „Detrital decomposition is a function of temperature and is allowed to occur when oxygen is zero. Even though nitrification and denitrification are not explicitly included in the model and the global nitrogen budget is kept constant (Oke et al., 2013), this formulation emulates the effect of denitrification."*

We have made the suggested changes, although phytoplankton and zooplankton are also prognostic tracers.

The paragraph now reads:

"WOMBAT is a nutrient-phytoplankton-zooplankton-detritus (NPZD) model (Oke et al., 2013; Law et al., 2017; Ziehn et al., 2020). It includes one functional type of phytoplankton and zooplankton. As prognostic tracers it simulates dissolved inorganic carbon (DIC), alkalinity (ALK), phosphate ($PO_4$), oxygen ($O_2$), and iron. The stoichiometry is fixed at a $C:N:P:O_2$ ratio of 106:16:1:-172. $CaCO_3$ export from the photic zone is set at ~8% of the organic carbon export. Detrital decomposition is a function of temperature and is allowed to occur when oxygen is zero. Even though nitrification and denitrification are not explicitly included in the model and the global nitrogen budget is kept constant (Oke et al., 2013), this formulation emulates the effect of denitrification. The dissolution of $CaCO_3$ occurs at a constant rate. All organic and inorganic particles reaching the bottom are remineralized, given that ACCESS-ESM1.5 does not include burial of sediments."

*Line 84: „All organic and inorganic particles reaching the bottom are remineralized, given that ACCESS-ESM1.5 does not include burial of sediments". Am I right in my assumption that the fact that carbon does not exit the ocean system at the lower boundary, but is instead fed back to the ocean via remineralization, may have an impact on simulated oxygenation of sea water? Please kindly refute or confirm my assumption.*

Yes, the Reviewer is correct.

*If you can confirm my assumption, then please explain (e.g. in your discussion) whether the effect of remineralization on oxygenation is relevant for the inferences that you derive from your study. In particular, is the effect comparable across different climates, so that the observed differences in oxygenation are fully attributable to a difference in climate states rather than a side-effect of the missing sediment module (that may be more (or less) important in dependence of the background climate state).*

In the real ocean, and in ocean models that include sediment modules, only a very small fraction of the tracers sinking to the ocean bottom is ultimately buried, while most are remineralised back into the water column. While we cannot quantify the effect of a missing sediment model on oxygen concentrations with the current modelling frame, we expect it to be very small. A potentially larger impact on oxygen patterns would be the nutrient exchange at the sediment-ocean interface, especially in oxygen-deprived regions (e.g. Niemeyer et al., 2017), but these processes are not included in any of the (rare) global Earth System Models with sediment components.

*Line 86: Since Eyring et al. (2016) describe the CMIP6 settings, maybe reformulate: „The pre- industrial 1850 equilibrium simulation (PI) is integrated following the CMIP6 protocol (Eyring et al., 2016) with the exception of using the CMIP5 solar constant (1365.65 W.m−2)."*

This has been amended.

*Line 107: Am I right that a „N/S" should be added after „40°"?*

We added "N/S" after "40º".

*Line 109: Also here please provide the reference(s) (unless Table 1 now clearly lists them).*

We have now added the references to the caption of Table 1. The caption reads now:

"**Table 1. Experimental set-up. LIG boundary conditions follow PMIP4 protocol (Otto-Bliesner et al., 2017). MIS 9e boundary conditions are based on Berger (1978) for orbital parameters, and peak concentrations for methane (Loulergue et al., 2008), carbon dioxide (Bereiter et al., 2015), and nitrous oxide (Schilt et al., 2010)**."

*Line 123-124: „This weakening of deep-ocean convection is mainly due to reduced sea-ice formation" - differences in sea ice formation will impact deep-ocean convection both with regard to the amount of brine rejection and the intensitiy of atmosphere-ocean coupling. Furthermore, presence or absence of sea ice will likely also modulate the atmosphere-ocean gas exchange. Can it be quantified or estimated which of these effects is more relevant?*

No, this is unfortunately not possible in a fully coupled ESM.

*Line 126: Please define Apparent Oxygen Utilisation. I am not sure whether all readers are familiar with this metric.*

This is now done in the Methods section, please see our reply above.

*Line 130: „with the increase in export production and higher remineralisation rates, and secondarily caused by the temperature-depended solubility"*

This has been amended.

*Line 133: „Figure 2a and d show that"*

This has been fixed.

*Line 144: Split the long sentence: „subtropical saline waters. Both are" Line 145-146: use the abbreviation NADW*

This was changed.

*Line 168: „at the surface (not shown), that is linked to lower primary productivity and export production"*

*Changed to:*

**"... at the surface (not shown), causing lower primary productivity and export production"**

*Line 180: Has significance been tested? If so consider to show this in the figures, e.g. via hatching?*

This was also flagged by Reviewer 1. We have now replotted all figures showing anomalies in a way that clearly highlights regions that are not significant.

*Line 190: For clarity, consider to move the statement „in the LIG simulation" to the start of the sentence.*

This was done.

*Line 215: Add commas for clarity of the formulation: „in the MIS9e simulation, dropping to 52.4 mmol m−3,"*

We added commas.

*Line 223: add a space: „128 ka BP", same at line 224 (and potentially at other locations)*

This was amended and checked for consistency throughout the text.

*Line 289: try to avoid the word segmenta of MIS 9e being separated by line breaks (maybe do not use a space in the term, across your manuscript?)*

This has been fixed throughout the text.

*Line 302: This region remains*

Fixed.

*Line 308: please check whether „dependent on" or „dependent of" is grammatically correct here (I am not native English speaker)*

"Dependent on" or "dependent upon" is correct. This has been fixed. Thank you.

*Line 312: long equilibration times?*

Fixed.

*Line 317-319: Is your statement regarding higher sensitivity to insolation than to greenhouse gases robust if also taking into account the differences in oxygenation found between interglacials in the Mediterranean? I think that a strict discrimination between different sensitivties and estimation of their relative strength would be only possible based on simulations that include separations of forcing, am I right?*

Yes, the reviewer is correct. A strict discrimination between different forcings would only be possible with additional sensitivity simulations. We believe, however, that our statement is correct, given that the large-scale circulation and oxygenation in the MIS 9e and LIG simulations are very similar and they are both very different from PI. The differences in the Mediterranean Sea between MIS 9e and LIG are mainly due to a combination of differences in monsoon strengths (which are mostly due to orbital parameters) and equilibration times.

*Line 330: „and 63% is hypoxic"*

Fixed.

*Line 335: consider the spelling of skillful, at least at one other location you spell it with one „l"*

Skillful is spelled with two l in American English and with one l in British/Australian English. We have now adopted British spelling of skilful throughout the text.

*Specific comments to Figures (beyond font size, which represents a problem across the manuscript) and Tables:*

*Table 1: Please provide the reference to the parameter values provided, in particular orbital parameter solution and reference for greenhouse gases. While on may guess that Otto-Bliesner et al. (2017) is the reference for PI and LIG, providing the citation is particularly important for MIS 9e where there is no obvious reference that comes to my mind. Such information may also be used to make the table heading a bit more informative. Regarding footnote 1: How, if at all, does erroneous forcing affect results? It is appreciated that such information is conveyed to the reader. Nevertheless, a bit of evaluation on the potential impacts (or the absence of such) may be of interest to the readers.*

We have now amended the caption of Table 1 (see above).

The first 372 years of our LIG simulation were accidentally integrated with PI greenhouse gas concentrations (and LIG orbital parameters). This does not affect the quasi-equilibrium state that is analysed here as the simulation was then equilibrated for a further 1450 years with the correct boundary conditions.

We have now clarified the footnote of table 1:

**"The first 372 years have erroneous forcing and were integrated with PI greenhouse gas concentrations."**

*Figure 2: Please define AOU in the caption (also in other Figure captions, where needed). Contour lines and contour line labels carry little information for me since I cannot read them.*

We have now spelled out AOU in all Figure captions and added a description of AOU to the Methods section (see above).

*Figure 4: Differences in black contour lines (solid, dotted, dashed) not visible to me. Furthermore, I am not sure about usage of footnotes in figure captions. I would avoid them and rather implement the information directly as caption text.*

These contour lines are almost identical and therefore only visible when zooming in (we still need to show them though). This is stated in the text (this Figure is now Figure 7):

**"The extent and intensity of OMZs is very similar between the three simulations at 300 m depth (Figure 7a and c)."**

The figure caption has been changed to:

**"$O_2$ concentration and anomalies at 300 m depth in hypoxic zones (left) and vertical minimum of $O_2$ concentration and anomalies in hypoxic zones (right) in (a, b) LIG - PI, (c, d) MIS 9e - PI, (e, f) PI and (g, h) World Ocean Atlas (WOA, 1965-2022). Black contour lines in subplots a-d indicate the 62 mmol·m$^{-3}$ isolines for PI (solid), LIG (dotted), and MIS 9e (dashed). Hypoxic zones are defined as zones where $O_2$ concentration at 300 m is below 62 mmol·m$^{-3}$ for (a, c, e, g) and where the vertical minimum of $O_2$ concentration is below 62 mmol·m$^{-3}$ for (b, d, f, h)."**

*Figure A1: Show hovmöller plots with calendar corrected data, unless there is a good reason to use the modern calendar.*

Our Hovmöller plot has an x axis that shows the day of the year, with each day weighted equally. When changing this to months (either for present day calendar or LIG calendar), as often done in publications, the plot becomes distorted and shorter months are over-represented. This is particularly true for the LIG adjusted calendar which has a larger range in month' lengths than the present day calendar. Our Hovmöller plot therefore shows neither modern day nor LIG calendar - just the 365 days of the year.

*Figure A2: It is difficult for me to extract any meaningful information from the very small legends, boxes, and thin lines. Keeping this in mind, it is also difficult for me to identify any interpolated data that you refer to in the caption.*

Figure A2 (new Figure 1) has been replotted.

*Figure A5: Can you explain the artifact at about 20°N in the AMOC (left column)? My first guess would have been the Strait of Gibraltar, but fluxes across the gateway are probably too small and the location of the artifact also does not fit.*

We would like to thank the Reviewer for catching this. This artifact was due to a small inconsistency in the mask used for the Atlantic Ocean. This has now been fixed.

*Figure A6: There is stronger AMOC in LIG and MIS9e (Fig. A5) but the mixed layer depths are smaller in these simulations than they are in PI; i.e., there is a weaker link between AMOC strength and mixed layer*

*depth in MIS9e and LIG. Does this difference in dynamics contribute to your findings regarding oxygenation?*

This is actually not quite correct, the mixed layer depths are not smaller in these simulations than they are in PI. When looking at Figure A6 (new Figure A3), one can see that a new convection site is active in the Labrador Sea in the MIS 9e and LIG runs. This new convection site contributes colder water to NADW and is responsible for the increase in ventilation. All other convection sites remain active in all three simulations, with the convection site south of Spitsbergen being weaker in the MIS9e and LIG runs compared to PI. We assume that the Reviewer was referring to that site? Overall, the new convection site makes up for this weakening.

*Furthermore, you mention in the main text that changes in deep mixing are linked to different sea ice conditions. So, can the weakened deep convection in the Barents Sea near Svalbard be explained by less autumn-to-winter sea ice formation and the related reduced brine rejection?*

The Reviewer is referring to following sentence in "Section 3.1.1 Antarctic Bottom Water":

**"This weakening of deep-ocean convection is mainly due to reduced sea-ice formation (Yeung et al., 2024; Choudhury et al., 2022) and leads to warmer, less ventilated, and therefore less oxygenated AABW."**

This sentence referred to convection in the Southern Hemisphere, which was analysed in detail in our group in earlier publications. We have not yet analysed the regional dynamics in the North Atlantic for these simulations, and it would be out of scope to do so for this manuscript.

*Figure A8: Would it make sense to also show the PMOC to understand the links between deep ventilation and meridional overturning?*

These rather small changes in ventilation unfortunately do not show up on stream function plots of the Pacific Ocean. That's why we opted to show the changes based on changes in mixed layer depth (old Figure A8, new Figure A5) and water age (old Figure A9, new Figure A6).

*Figure A9: Is there a reason why you compare in a and b different depths with each other (250 m vs 50 m)?*

We apologize, that was a typo. It was meant to read 250 m for both subplots.

Done.

*Eyring, V., Bony, S., Meehl, G. A., Senior, C. A., Stevens, B., Stouffer, R. J., and Taylor, K. E.: Overview of the Coupled Model Intercomparison Project Phase 6 (CMIP6) experimental design and organization, Geosci. Model Dev., 9, 1937–1958, https://doi.org/10.5194/gmd-9-1937-2016, 2016.*

Melki, T., Kallel, N., Fontugne, M.: The nature of transitions from dry to wet condition during sapropel events in the Eastern Mediterranean Sea, Palaeogeography, Palaeoclimatology, Palaeoecology, 291, 267-285.

Niemeyer, D., Kemena, T.P., Meissner, K.J. and A. Oschlies, 2017: A model study of warming-induced phosphorus-oxygen feedbacks in open-ocean oxygen minimum zones on millennial timescales. Earth System Dynamics, 8, 357-367.

*Otto-Bliesner, B. L., Braconnot, P., Harrison, S. P., Lunt, D. J., Abe-Ouchi, A., Albani, S., Bartlein, P. J., Capron, E., Carlson, A. E., Dutton, A., Fischer, H., Goelzer, H., Govin, A., Haywood, A., Joos, F., LeGrande, A. N., Lipscomb, W. H., Lohmann, G., Mahowald, N., Nehrbass-Ahles, C., Pausata, F. S. R., Peterschmitt, J.-Y., Phipps, S. J., Renssen, H., and Zhang, Q.: The PMIP4 contribution to CMIP6 – Part 2: Two interglacials, scientific objective and experimental design for Holocene and Last Interglacial simulations, Geosci. Model Dev., 10, 3979–4003, https://doi.org/10.5194/gmd-10-3979- 2017, 2017.*

*Duteil, O., Koeve, W., Oschlies, A., Bianchi, D., Galbraith, E., Kriest, I., and Matear, R.: A novel estimate of ocean oxygen utilisation points to a reduced rate of respiration in the ocean interior, Biogeosciences, 10, 7723–7738, https://doi.org/10.5194/bg-10-7723-2013, 2013.*

We would like to thank the Reviewer again for the extremely exhaustive and helpful review that helped us improve the paper.

---

## Author Response (AR2)

Dear Dr. Zhang,

We have now addressed the remaining technical corrections and resubmitted the paper. Please find our responses to Referee #3's suggestions below.

We would like to thank you again for handling this submission.

Kind regards,

Bartholomé Duboc and co-authors

**Response to Report #1 (Referee #3) – Technical corrections**

We would like to thank Referee #3 for their exhaustive and helpful review. We have replicated the Reviewer's comments below in blue and italics. Our responses to each of the comments are in black.Thank you for your time and expertise!

*Figure 1: „[...] we show therefore interpolations (long dashed lines) over this time span."*
The caption has been corrected.

*Figure 2: I note that abbreviations differ between plot labels and figure caption (ΔTair vs. SAT and ΔTsst vs. SST); I suggest to harmonize terminology.*

Indeed, we have now harmonized the terminology by keeping SST and SAT in both labels and caption.

*Figure 6: Please make sure that the text or the caption conveys the meaning / origin of white-colored areas, in particular for some coastal regions and gateways. I assume there is a lack of data due to ocean bathymetry being lower than 250 m (e.g. Bering Strait) or specific regions not being actively simulated in the model (e.g. Caspian Sea), am I right?*

The Reviewer is correct and we have amended the figure caption accordingly.

*Line 214: observe the dot that jumped to the next page*

We are not sure what the Reviewer is referring to here. We have double checked the revised manuscript and have not seen any displaced dot.

*Figure 10 d: Should the axis label read „net freshwater budget" instead of „net water budget"?*

The axis label has been fixed.

*Figure A7: Please provide a scale for the vectors.*
The vector scale has been added in all the subplots. We have also reduced the size of the arrows in the last two plots to make them easier to read.

*Figure A8: Should „Full fields" be replaced by „Absolute values"?*

No, the Reviewer is mistaken here, it should be "full fields".